evolution/genetics/genomics

Neanderthal, Denisovan, introgression, mutation rate, heterozygosity, out of Africa

**Author for correspondence:**
William Amos
e-mail: w.amos@zoo.cam.ac.uk

# Correlated and geographically predictable Neanderthal and Denisovan legacies are difficult to reconcile with a simple model based on inter-breeding

## William Amos

Department of Zoology, Cambridge University, Downing Street, Cambridge CB2 3EJ, UK

WA, 0000-0002-0971-9914

Although the presence of archaic hominin legacies in humans is taken for granted, little attention has been given as to how the data fit with how humans colonized the world. Here, I show that Neanderthal and Denisovan legacies are strongly correlated and that inferred legacy size, like heterozygosity, exhibits a strong correlation with distance from Africa. Simulations confirm that, once created, legacy size is extremely stable: it may reduce through admixture with lower legacy populations but cannot increase significantly through neutral drift. Consequently, populations carrying the highest legacies are likely to be those whose ancestors inter-bred most with archaics. However, the populations with the highest legacies are globally scattered and are unified, not by having origins within the known Neanderthal range, but instead by living in locations that lie furthest from Africa. Furthermore, the Simons Genome Diversity Project data reveal two distinct correlations between Neanderthal and Denisovan legacies, one that starts in North Africa and increases west to east across Eurasia and into some parts of Oceania, and a second, much steeper trend that starts in Africa, peaking with the San and Ju/'hoansi and which, if extrapolated, predicts the large inferred legacies of both archaics found in Oceania/Australia. Similar 'double' trends are observed for the introgression statistic $f_4$ in a second large dataset published by Qin and Stoneking (Qin & Stoneking 2015 *Mol. Biol. Evol.* **32**, 2665–2674 (doi:10.1093/molbev/msv141)). These trends appear at odds with simple models of how introgression occurred though more complicated patterns of

introgression could potentially generate better fits. Moreover, substituting archaic genomes with those of great apes yields similar but biologically impossible signals of introgression, suggesting that the signals these metrics capture arise within humans and are largely independent of the test group. Interestingly, the data do appear to fit a speculative model in which the loss of diversity that occurred when humans moved further from Africa created a gradient in heterozygosity that in turn progressively reduced mutation rate such that populations furthest from Africa have diverged less from our common ancestor and hence from the archaics. In this light, the two distinct trends could be interpreted in terms of two 'out of Africa' events, an early one ending in Oceania and Australia and a later one that colonized Eurasia and the Americas.

# 1. Introduction

It is now largely accepted as a fact that humans inter-bred widely with other archaic hominins such as Neanderthals and Denisovans [1–8] apparently more or less whenever the species' ranges overlapped [8–10]. Evidence for this can be divided broadly into three main forms. Most direct is the discovery of skeletons of individuals whose DNA appears to reflect two different hominins in their very recent ancestry [10,11]. Second, using aligned genomes, it is possible to estimate which of two humans on average across the genome share more bases with an archaic hominin than the other [12,13]. Finally, the presence of significant clusters of derived archaic bases in humans can be used to infer the presence of introgressed haplotypes [6,14,15].

Arguably the most accessible and, therefore, most widely used method to quantify the total, genome-wide archaic legacy is the so-called ABBA–BABA test, usually expressed as the statistic $D$ [5,12,16,17]. The ABBA–BABA test is typically applied to biallelic sites in a four-way alignment ($P1,P2,P3,P4$) where the chimpanzee ($P4$, always allele 'A') and the archaic ($P3$, always allele 'B') differ. Informative sites are those where the two humans, $P1$ and $P2$, also differ, generating either an 'ABBA' or a 'BABA' conformation. $D$ is then calculated as the normalized difference in counts, (ABBA – BABA)/(ABBA + BABA), and varies between −1 and 1. For the classic comparison between an African and a European with archaic = Neanderthal, $D$ is approximately 5%. If the mutation rate is constant and back-mutations are rare enough to be ignored, and if humans and Neanderthals diverged approximately 300 000 years ago, this equates to around 1.5–2% introgressed DNA in Europeans [5,18]. However, such estimates are acutely sensitive to the timing of the split. Earlier split times [19,20] that fit better with the archaeology [21] would imply much smaller legacies for the same $D$ [12].

While the first reports of inter-breeding between archaics and humans focused on Neanderthals [1,5], the discovery and sequencing of a second archaic hominin, the Denisovan [22,23], led to a more complicated picture in which gene flow into humans was inferred from both archaics [7] and perhaps other, as yet unidentified species/lineages [9]. With more global human genomes being completed came more refined estimates for how the sizes of these legacies varied globally [4,24–26]. Trends have been identified for Neanderthal legacies to increase west to east across Eurasia [27], for a large spike in Denisovan contribution in Papuans [4,23] and aboriginal Australians [25] and for limited archaic contributions in some African populations [28]. These complicated trends raise issues about how introgressed Neanderthal, Denisovan and potentially other archaic fragments can be distinguished, particularly using generic statistics such as $D$. One solution is to focus on alleles that are derived within each different archaic and use these alleles to construct joint landscapes [7].

Despite a high level of general acceptance, the idea archaic legacies are near-ubiquitous in humans is not without issue. As yet, convincingly archaic mitochondrial DNA, X chromosomes or Y chromosomes have yet to be found in humans. While this absence might be the result of purifying selection, this implies that hybrid individuals would have appreciably reduced fitness, presenting a barrier to gene flow. More directly, hybridization since the out of Africa event around 70 000 years ago [29] would result in legacies where most introgressed fragments are rare and hence heterozygous. I have tested this prediction by using a form of conditioned $D$ where sites are excluded/included depending on whether they are heterozygous/homozygous in one of the two humans being compared [30]. I found that the excess base sharing between non-Africans and archaics captured by $D$ is driven almost entirely by heterozygous sites in Africa acting to *increase* African–Neanderthal divergence rather than introgressed fragments in non-Africans acting to *decrease* non-African–Neanderthal divergence: the exact opposite of what is expected under the classical inter-breeding hypothesis.

Non-zero $D$ can only arise in two ways: through introgression and through mutation rate variation between lineages. Introgression acts to bring one human population closer to the archaic by increasing

shared ancestry. Mutation rate variation is routinely assumed to be absent but, if present, could allow one human population to diverge more than the other, both from the common ancestor of humans and, hence, from related lineages like Neanderthals. In fact, mutation rates do vary between human populations [25], with a large excess mutation rate seen in Africans when the data are filtered to remove variants that probably arose before the out of Africa event [31]. Moreover, across the genome, the excess mutation rate in Africa is strongly predicted by the amount of diversity lost [31], providing support for heterozygote instability (HI) [32,33], a relatively new hypothesis in which mutation rates are elevated in the vicinity of heterozygous sites. Further support for the idea that mutation rates vary comes from dramatic changes in the mutation spectrum, seen both between Africans and non-Africans and among different major geographical regions in Eurasia [34,35]. Changes in relative frequency cannot accrue unless different types of mutation occur at different rates. Interestingly, changes in the mutation spectrum have since been linked to variation in flanking sequence heterozygosity [36]. Taken together, this evidence suggests an alternative model in which the linear decline in heterozygosity with distance from Africa [37] generated a parallel decline in the mutation rate that in turn created the trend whereby affinity to archaics increases from west to east.

One way to help distinguish between introgression and mutation rate variation as mechanisms capable of causing variation in levels of base sharing between different human populations and archaics may be to examine the extent to which different legacies are correlated. The introgression model predicts that legacies will tend to reflect where and when inter-breeding occurred. For example, the large peak of inferred Denisovan legacy in Papuans [4,23,26] and much lower levels elsewhere suggests that inter-breeding with Denisovans occurred mainly in East Asia/Oceania. Equally, the ubiquity of the inferred Neanderthal legacy is difficult to explain unless most inter-breeding occurred in the Levant [3,38], soon after humans migrated out of Africa. The two legacies are, therefore, expected to be un- or even negatively correlated. By contrast, if inferred legacies are artefacts linked to global variation in human mutation rate, the two legacies should tend to covary.

Here, I test predictions from these two competing hypotheses through analysis of the 1000 genomes data, through a reanalysis of two large published studies and through coalescent simulations. Specifically, I ask: (i) whether signals of introgression for the two archaics are correlated, as expected under mutation slowdown, or show appreciable independence, as expected if inter-breeding between the two archaics occurred at different geographical locations; (ii) whether signals of introgression are strongly predicted by distance from Africa, as expected if heterozygosity modulates mutation rate, or links mainly to sites where inter-breeding most likely occurred; (iii) whether substitution of archaic hominins for great apes results in a zero signal, as expected under introgression, or still generates a signal, as expected if the signal originates largely by processes within humans, the non-humans acting merely as a marker of the ancestral state. I find that metrics used to infer legacies are strongly correlated between Neanderthals and Denisovans in a way that appears difficult to explain by a simple model based mainly on introgression, that introgression statistics are strongly correlated with distance from Africa and that three great apes all yield clear but biologically implausible signals of introgression.

## 2. Results

The classic $D(P1, P2, P3, P4)$ statistic is a simple measure that assumes 100% of the signal of introgression comes from the archaic genome included as P3. However, with more than one potential source of introgressed DNA, there can be overlapping signals due to bases shared between the different archaics. To separate these signals, Sankararaman *et al.* [7] introduce the measure $nd_{10}$ and $nd_{01}$, where they count bases that are derived in Neanderthals and Denisovans, respectively. Specifically, where introgression by both Neanderthals and Denisovans is possible, there are now five taxa to consider: $(P1,P2,P3,P4,P5)$, where $P1$ and $P2$ are the two humans, $P3 =$ Neanderthal, $P4 =$ Denisovan and $P5 =$ chimpanzee. Writing the two human alleles as X to indicate that they can be either A or B, putative derived Neanderthal alleles are written XXBAA (i.e. chimpanzee and Denisovan both ancestral), putative derived Denisovan alleles are written XXABA (Neanderthal and chimpanzee both ancestral) and putative ancestral archaic alleles are written XXBBA (both archaics share the same derived allele). If $P1$ is a panel of sub-Saharan Africans carrying no B alleles, $nd_{10}$ is calculated as the probability that $P2 =$ B in the second human population, calculated conservatively using only variants generated by transversion mutations [7]. $nd_{10}$ is, therefore, the average frequency of derived Neanderthal variants in a (usually) non-African population, conditional on these variants being absent in a panel of sub-Saharan individuals. $nd_{01}$ is the equivalent measure calculated for derived Denisovan alleles.

The original description of $nd$ uses '01' and '10' subscripts to denote Neanderthal and Denisovan signals, respectively, and these can be easy to misread. For added clarity, therefore, I will use $nd$ to refer to the measure in general terms and use $nd_{NEA}$ and $nd_{DEN}$ to refer its application to derived Neanderthal and Denisovan alleles, respectively. $nd$ is unusual in the way it seeks to minimize overlap between signals of introgression due to Neanderthal and those due to Denisovans, but at the same time, it suffers from an inherent bias. By conditioning on a panel of sub-Saharan Africans not carrying the derived archaic allele, it effectively prejudges the idea that archaic introgression into Africa never occurred and also, potentially, underestimates introgression into populations that are most closely related to Africans, particularly wherever there is any appreciable gene flow. This is because derived archaic variants found in the African panel will be automatically excluded. In reality, recent studies suggest introgression did occur into Africans [25,28]. Moreover, by focusing only on transversions, approximately two-thirds of the data are being discarded. Consequently, I decided to explore alternative versions of $nd_{NEA}$, with and without conditioning on the African state and comparing the uses of transitions with that of transversions.

## 2.1. The 1000 genomes data

Plots of $nd_{NEA}$ against $nd_{DEN}$ are given in figure 1, colour-coded by region of origin (Europe, red; East Asia, yellow; South Asia, blue; Africa, black; America, green). Plots based on transitions only (top row) and transversions only (bottom row) are essentially identical. However, conditioning on the derived archaic allele being absent from a panel of Africans has a huge impact. In the conditioned state (i.e. $nd_{10}$ *sensu* Sankararaman *et al.* [7,25]), the $nd_{NEA}$ and $nd_{DEN}$ values are positively correlated and increase from Africa through Europe, America and South Asia to East Asia. By contrast, there is a strong negative correlation when the requirement for no B alleles in Africa is relaxed. The reason for this negative correlation is unclear, though it may simply reflect the fact that, by defining states in terms of both archaics, similarity to one automatically means lack of similarity to the other. Note that in the conditioned state, the five African populations (by definition) have $nd = 0$ while the two admixed African samples (ACB/ASW) both have very low values: B alleles that are absent from five Africa populations are likely to be absent or at very low frequency in the other two African population samples.

Previous studies have reported that the estimated legacies for both Neanderthals and Denisovans are higher in the east than in the west [4,25,27] and this pattern is replicated here for $nd$. For Denisovans, this result is consistent with the high levels of introgression inferred in Oceania/Australia [4,23]. However, Neanderthal introgression is thought to have occurred soon after the out of Africa event, a scenario that appears at odds with higher legacies in the east. To address this issue, I calculated $nd$ values for each chromosome separately. Neutral drift should not change the mean legacy: even if the average change in inferred legacy is positive, equal numbers of chromosomes should exhibit west to east declines and increases. Similarly, natural selection should impact chromosomes differently, depending on the genes they carry. Since geographical origin is difficult to define for several 1000 genomes populations (e.g. south Asian groups sampled in the UK/US), I tested for consistency of signal across chromosomes by calculating the correlation coefficient for $nd$ values between each chromosome and the average for all other chromosomes (i.e. excluding the current chromosome). Chromosome 21 was omitted because lack of good alignments across all taxa mean that it contributes 20–50-fold fewer sites even than other, equally small chromosomes. Strong correlations are observed for both Neanderthals (mean $r = 0.95$ (transitions) and 0.94 (transversions)) and for Denisovans (mean $r = 0.84$ (transitions) and 0.82 (transversions)). Such a consistent signal across the genome appears to preclude both selection and drift following a single dominant introgression event as mechanisms that could have created the observed rising west to east trends.

## 2.2. Simons Genome Diversity Project data

For a more extensive view on global trends, I turned to $nd$ values from the Simons Genome Diversity Project data published by Mallick *et al.* [25] for both Neanderthals and Denisovans for 300 genomes sampled from across the globe. Data from the whole dataset are plotted in figure 2. The data points form three main clusters. At the top is a group of brown points representing Australians and Papuans who both exhibit large $nd_{NEA}$ values and extremely large $nd_{DEN}$ values. Bottom right is a cluster comprising all other samples from outside Africa plus, arguably, the four North African samples (Algeria and Morocco). Finally, the remaining Africans (black) form a more complicated pattern. Most samples are zero for both measures, something that follows directly from the conditioning that alleles must be absent from a panel of sub-Saharan Africans, but there is also an unexpected, strong second

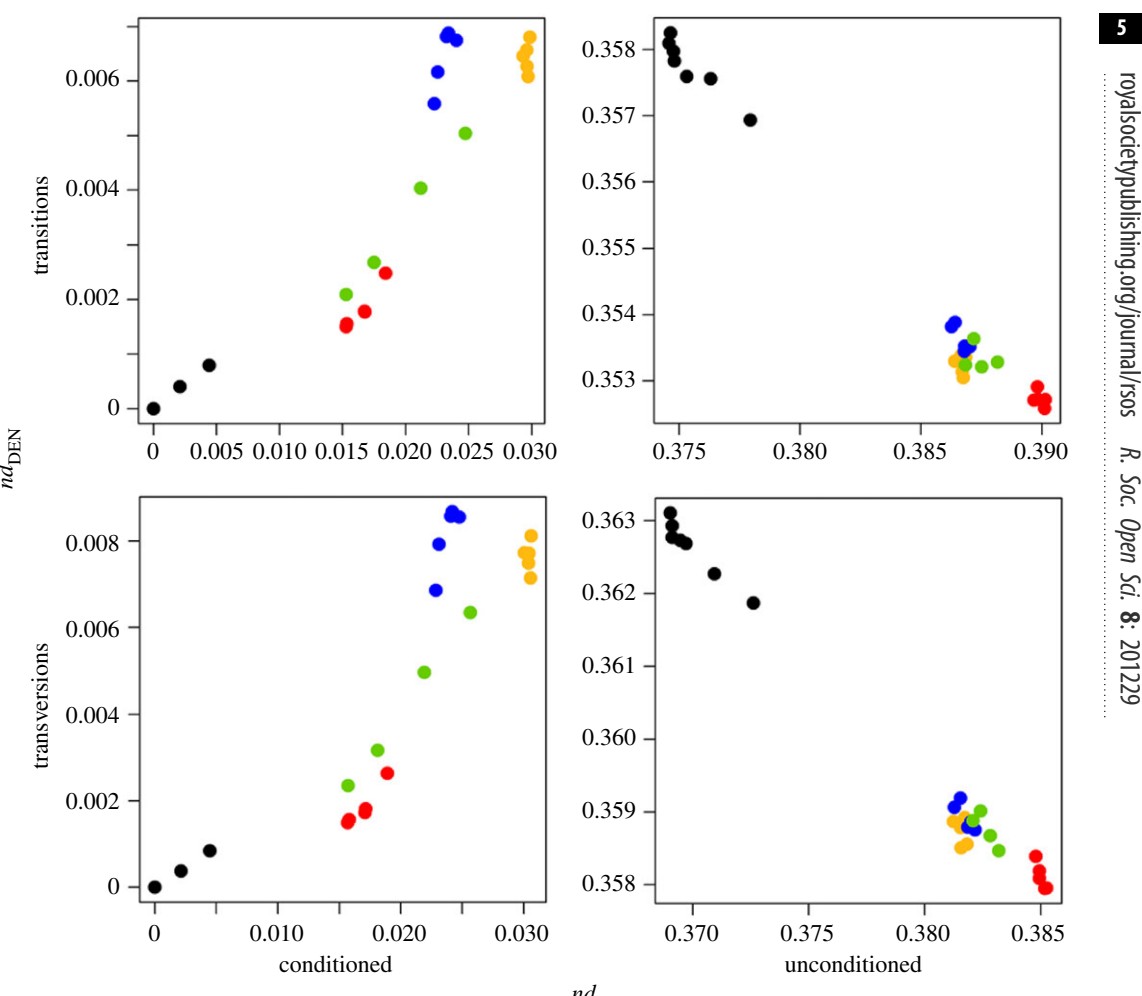

**Figure 1.** Correlations between the frequencies of derived Neanderthal and derived Denisovan alleles across the 1000 genomes populations. Derived allele frequencies are captured as *nd*, the average frequency of derived B alleles (both chimpanzee and the other archaic are A). Each analysis was repeated four times, partitioning the data according to whether the mutation creating the polymorphism was a transition or a transversion, and whether or not there was conditioning on the B allele being absent from the five non-admixed African populations. Data points for the 26 populations are colour-coded according to regional origin: Africa, black; Europe, red; South Asia, blue; East Asia, yellow; America, green.

trend within Africa that includes the San and Ju/'hoansi samples and appears to 'point' towards the Papuan/Australian cluster.

For a clearer view of what is happening in Eurasia, I replotted the data but this time excluding any data points with an $nd_{NEA}$ value below 0.01 or an $nd_{DEN}$ value above 0.02 (figure 3, essentially zooming in to the lower right portion of figure 2). As with the 1000 genomes data, a strong positive correlation exists between the two *nd* values ($r^2 = 0.44$, $N = 232$, $p < 2.2 \times 10^{-16}$). To expose any geographical trends, I next plotted land-only distance to East Africa against $nd_{NEA}$ (figure 4a, points colour-coded according to the corresponding $nd_{DEN}$ value) and against $nd_{DEN}$ (figure 4b, points colour-coded according to their corresponding $nd_{NEA}$ value). In both cases, a strong positive correlation is found, particularly for $nd_{NEA}$ where the greater number of derived bases reduces stochastic sampling noise. Not surprisingly, $nd_{DEN}$ and $nd_{NEA}$ are also strongly correlated with each other (figure 4c, points colour-coded by distance from Africa). Using the same data, I also fitted a multiple regression and found that $nd_{NEA}$ is predicted by $nd_{DEN}$ ($p < 2.2 \times 10^{-16}$) but that distance from Africa ($p = 1.4 \times 10^{-8}$) and the interaction term between these two predictors ($p = 0.0002$) are also highly significant.

## 2.3. A second global dataset and a second measure of introgression

The above analysis focuses heavily on *nd* which, although providing a useful way to split signals of introgression into those probably due to Neanderthals and those probably due to Denisovans, could

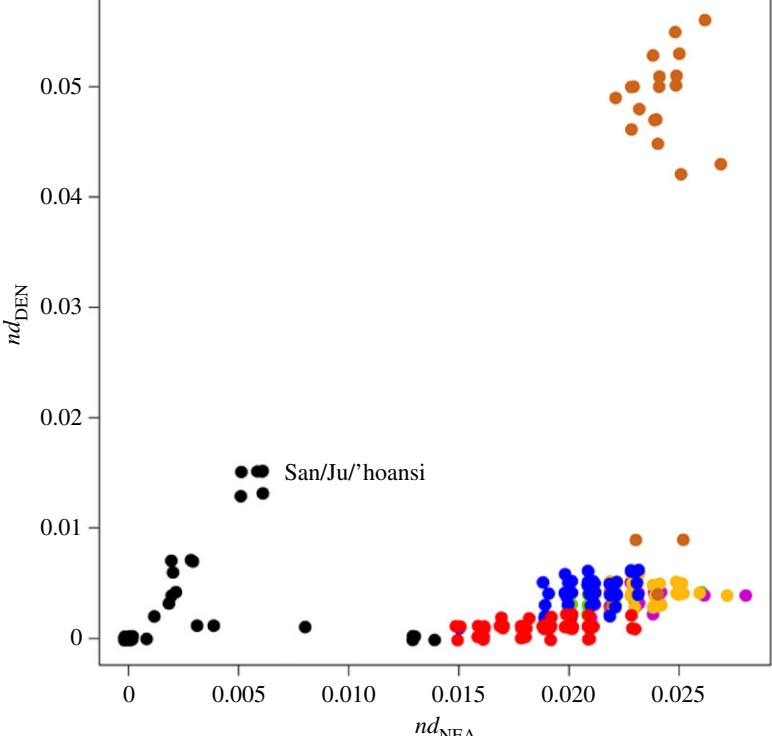

**Figure 2.** Correlation between $nd_{NEA}$ and $nd_{DEN}$ across the Simons Genome Diversity Project data. $nd$ values were taken directly from Mallick *et al.* Supplementary table 1 [25]. Populations are colour-coded according to the region of origin: Africa, black; West Eurasia, red; South Asia, blue; East Asia, yellow; America, green; Oceania/Australia, brown; Central Asia Russia, purple. Data are plotted with jitter. Since $nd$ is conditioned on the derived archaic allele being absent from a panel of sub-Saharan Africans, many African are zero for both measures.

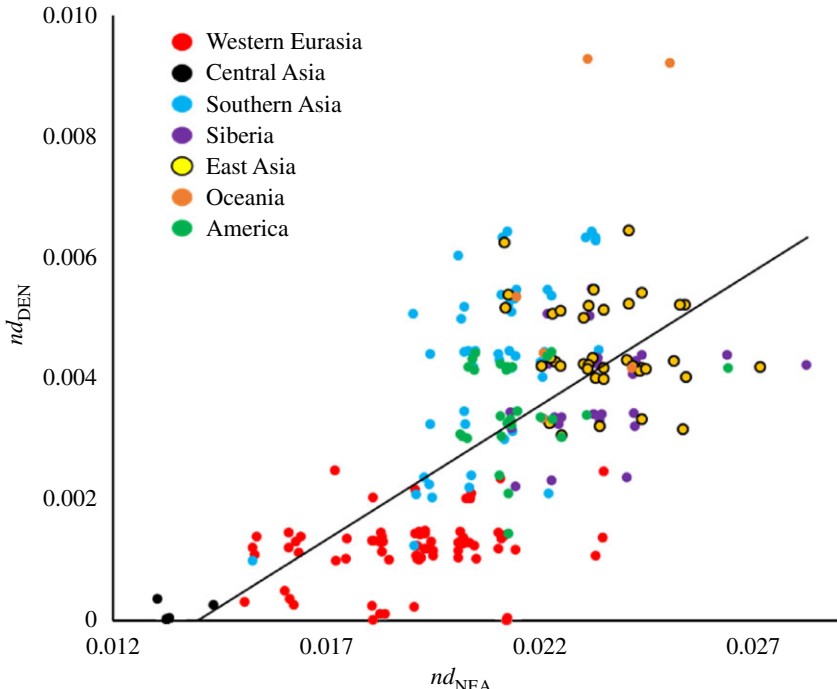

**Figure 3.** Correlation between $nd_{NEA}$ and $nd_{DEN}$ across the Eurasian samples from the Simons Genome Diversity Project data. $nd$ values were taken directly from Mallick *et al.* Supplementary table 1 [25]. This figure is the same as figure 2 except that African samples (apart from four North African samples) and samples with very high $nd_{DEN}$ values (Australia and Papua New Guinea) are excluded. This plot is a subset of the data plotted in figure 2. A strong positive correlation is observed ($r = 0.658$, $n = 231$, $p < 0.00001$).

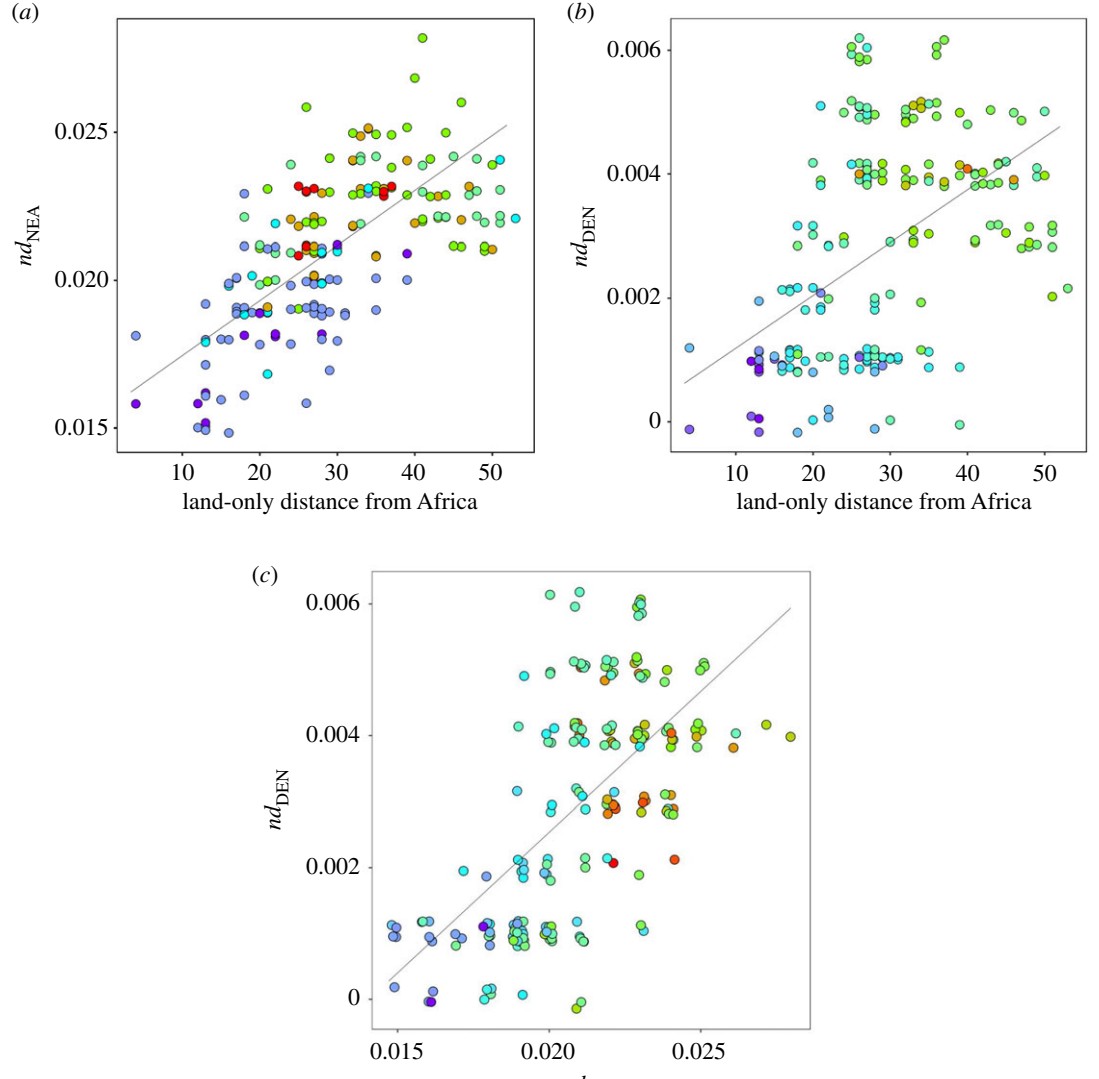

**Figure 4.** Relationship between distance from Africa, $nd_{NEA}$ and $nd_{DEN}$ across the Eurasian samples from the Simons Genome Diversity Project data. $nd$ values were taken directly from Mallick *et al.* Supplementary table 1 [25]. Land-only distances from Africa, taken as Addis Ababa, were calculated using the R package geoGraph. (*a*) How distance from Africa (arbitrary units) predicts $nd_{NEA}$, colour-coded according to equivalent $nd_{DEN}$ value, from blue (smallest values), through green to red (largest values). A strong positive correlation is observed ($r = 0.751$, $n = 294$, $p < 0.00001$). (*b*) How distance from Africa (arbitrary units) predicts $nd_{DEN}$, colour-coded according to the equivalent $nd_{NEA}$ value, from blue (smallest values), through green to red (largest values). A strong positive correlation is observed ($r = 0.663$, $n = 294$, $p < 0.00001$). (*c*) How $nd_{NEA}$ predicts $nd_{DEN}$, colour-coded according to distance from Africa, colour-coded from blue (nearest to Africa), through green to red (furthest from Africa). Again, a strong positive correlation is observed ($r = 0.669$, $n = 294$, $p < 0.00001$).

be seen as somewhat unusual in the way it conditions on derived alleles being absent from a panel of sub-Saharan Africans. For a comparison, I turned to a second large published dataset from Qin & Stoneking [4], who use a second widely used statistic $f_4$ [13,16]. Like $D$, $f_4$ is calculated from four taxa. Here, the taxa are split into two pairs (*P1, P2; P3, P4*) and a correlation sought between the paired allele frequency differences, *P1–P2* and *P3–P4*. The two differences are expected to be uncorrelated unless introgression has occurred, in which case a correlation can be generated.

Qin & Stoneking present several different $f_4$ statistics, including $f_4$(*Yoruba, X; Neanderthal, Denisovan*), where *X* is any of a wide range of Eurasia, American and Oceanian populations (their Supplementary table 4, [4]). In their words 'An excess of allele sharing with Denisovan yields positive values while an excess with Neanderthal yields negative values'. This is perhaps surprising, given that all values in their table are negative even though Europeans are believed to harbour far more Neanderthal DNA than Denisovan [7,18] while Oceanians should carry far more Denisovan DNA than Neanderthal

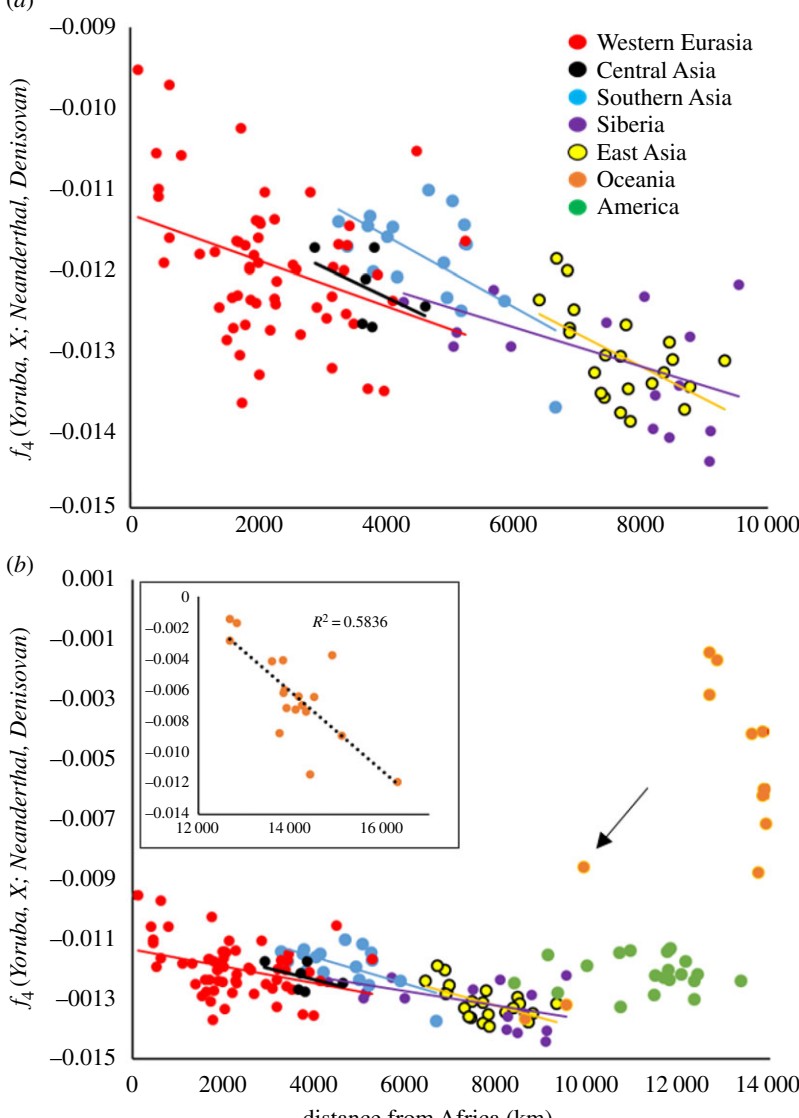

**Figure 5.** Global patterns of inferred archaic introgression, measured using the $f_4$ statistic. All data are taken directly from Qin and Stoneking, Supplementary table 4 [4] and relate to the measure $f_4$(Yoruba, X; Neanderthal, Denisovan) where more negative values indicate a higher affinity to Neanderthals and more positive (less negative) values indicate a relatively higher Denisovan proportion. Distances from Africa are calculated as Great Circle distances from Cairo, taken as 30° N, 31° E. Great Circle distances were used as a pragmatic way to deal with the large number of Oceanic populations whose routes of colonization remain unclear. Cairo was used as the origin so as to minimize the impact of presumptions about the location of the origin of the out of Africa event(s). (a) Data only for Eurasian populations (Western Eurasia, red; Siberia, purple; South Asia, blue; Central Asia, black; East Asia, yellow) with linear regression lines fitted separately to each group. (b) The same data but with the addition of Oceania (orange) and America (green). The unusual Mamanwa population is arrowed. The main Oceania cluster has a trend that is too steep to show clearly on the same scale as the remaining data so is portrayed separately as an inset.

[4,23]. Nonetheless, I began by asking whether their Eurasian data also exhibit a dependence on distance from Africa (figure 5a), finding a strong negative correlation ($r = 0.389$, $n = 121$, $p = 0.00001$). Moreover, all individual population groups yield slopes that do not differ significantly from each other (glm: $f_4 \sim$ distance × region; distance, $p < 2 \times 10^{-16}$; region, $p = 0.0123$; interaction term, $p = 0.88$, $n = 121$). Adding in the Americans and Oceanians creates a dramatic contrast (figure 5b). The Americans (green) form a loose cluster with no trend, possibly/probably because of varying levels of admixture. Most Oceanians form a distinct cluster with high (less negative) values and a significant, much steeper negative trend ($r = 0.764$, $n = 19$, $p = 0.00014$). Two Oceanian populations fall within the main Eurasian trend (Borneo and Semende) while one, the Mamanwa (arrowed), represent a major outlier (as also

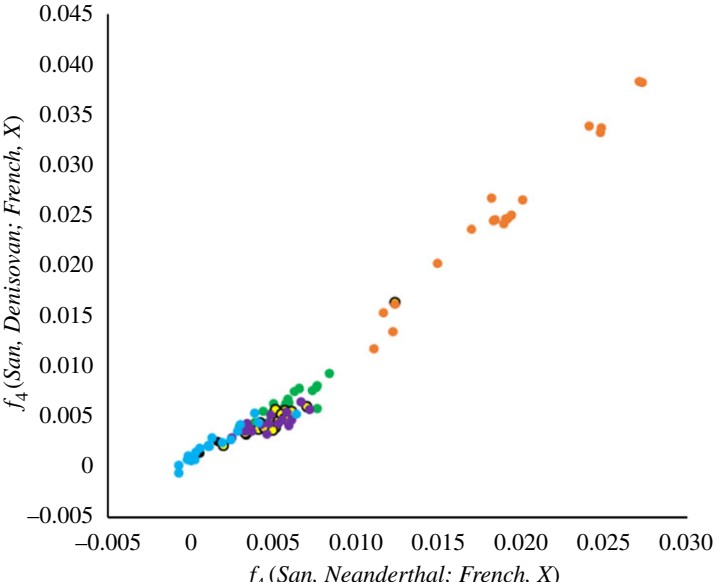

**Figure 6.** Correlated signals of introgression using the $f_4$ statistic. To explore the extent to which signals interpreted as introgression are correlated when using the $f_4$ statistic, I again used data from the large Qin and Stoneking study, focusing on the paired $f_4$ values in their Supplementary table 3. The main plot is similar to a number of directly equivalent plots in their paper, with $f_4$(San, Neanderthal; French, X) on the X-axis and $f_4$(San, Denisovan; French, X) on the Y-axis for the Eastern populations Oceania, East Asia, Central Asia, South Asia and America, colour-coded as in figure 5.

noted by Qin & Stoneking [4]). Interestingly, if they are part of the general Eurasian trend, the Mamanwa $f_4$ value would suggest a location inside Africa.

Qin & Stoneking [4] also publish data for paired $f_4$ values of the form $f_4$(San, Archaic; French, X), where *Archaic* = Neanderthal or Denisovan and X = various Eastern population groups: America, East Asia, South Asia, Oceania and Central Asia (see their Supplementary table 3). Plotting the corresponding pairs of $f_4$ values against each other (figure 6) reveals a strikingly strong positive relationship (as also seen in a number of related plots in their paper). This plot is troubling because any given significantly non-zero value would be interpreted as a measure of archaic introgression, implying that legacies across all these populations are extremely highly correlated, even more so than seen for the *nd* statistic. This raises the possibility that background correlations may be present even in the absence of introgression.

To test the possibility that background signals are present even without archaic introgression, I replicated their analysis using the 1000 genomes data, but this time including taxa where introgression would be biologically impossible. Specifically, I calculated $f_4$(ESN, Taxon; GBR, X) where: ESN is one of the African populations; Taxon = Neanderthal, Denisovan, chimpanzee, gorilla or orangutan; and X = any of the other 1000 genomes populations. Plotting $f_4$(ESN, Neanderthal; GBR, X) against each of the other taxa reveals similar strong correlations but with different intercepts (figure 7), with taxon = Denisovan (black) having an intercept ∼ 0 and the great apes each yielding increasingly positive intercepts (in order, chimpanzee, blue; gorilla, red; orangutan, orange). Furthermore, taking one representative comparison, $f_4$(ESN, Taxon; GBR, CHB), CHB being Chinese from Beijing, and calculating paired values for each autosomal megabase across the genome reveals that the $f_4$(ESN, Neanderthal; GBR, CHB) values are all strongly correlated with their equivalents in all four other taxa (figure 8), showing that the correlated signals apply effectively to the entire genome.

## 2.4. Simulations

To assess the plausibility of different patterns that could be caused by introgression followed by neutral drift, I conducted coalescent simulations using the program ms [39]. I simulated a scenario where Neanderthals and Denisovans are sister taxa, where non-Africans suffer an out of Africa bottleneck and receive a pulse of Neanderthal DNA before a series of five founder events generates a series of six non-African populations linked by bidirectional migration between adjacent populations. Immediately after founding, the last non-African population receives a pulse of Denisovan DNA (figure 9). Migration rates were varied from 1 to 50

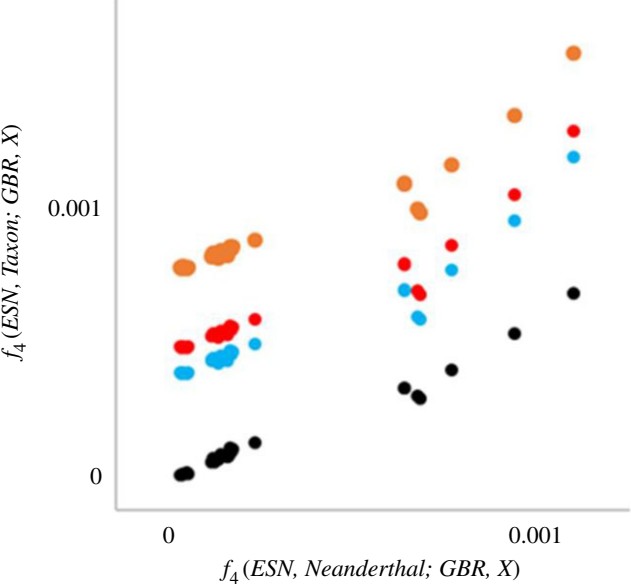

**Figure 7.** Performance of $f_4$ statistic in the presence of taxa where introgression is biologically implausible. This plot emulates figure 6 but using the 1000 genomes data. I chose Esan (ESN) as my African population and Britain (GBR) as my European population, and calculated $f_4$(ESN, taxon; GBR, X), where X is one of the other 26 1000 genomes populations and taxon is a non-human outgroup: the Denisovan (black), chimpanzee (blue), gorilla (red) or orangutan (orange). In the resulting plot, each 1000 genomes population contributes a data point to each series, the X-axis is always $f_4$(ESN, Neanderthal; GBR, X) and the Y-axis is $f_4$ for each of the other four taxa.

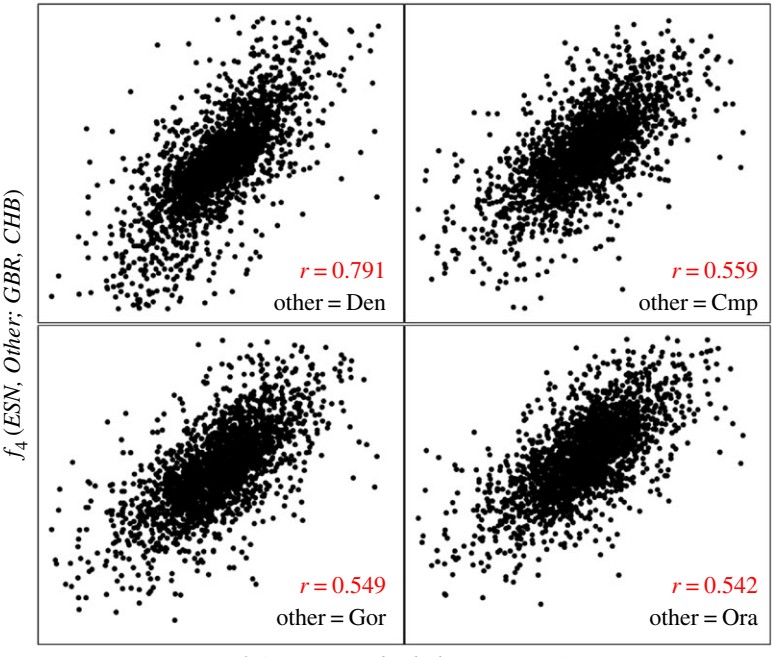

**Figure 8.** Consistency of biologically implausible signals of introgression across the genome. I chose one representative comparison from the 1000 genomes data, featuring one European (GBR, Britain) and one East Asian (CHB, Chinese from Beijing) and calculated the five non-human $f_4$ values described in figure 7 for each non-overlapping megabase in the autosomal genome. For comparability, each plot is deliberately made using the same scale ($-0.004$ to $0.004$ on both axes). This excludes modest numbers of sometimes outliers, typically 3–4% and always less than 5% of data points (mostly windows containing much reduced numbers of informative sites). Almost all the large outliers are associated with comparisons involving great apes, and this accounts for why the Denisovan correlation is so much stronger despite appearing very similar. All correlations are highly significant ($r > 0.5$, $n = 2330$, $p \ll 0.00001$). Correlations for other comparisons of the form $f_4$(ESN, Neanderthal; GBR, X) versus $f_4$(ESN, great ape; GBR, X) all yield very similar values. When X = Europe or Africa, all correlations become somewhat weaker ($r \sim 0.3$ for great apes) or marginally stronger ($r \sim 0.59$ for great apes), respectively.

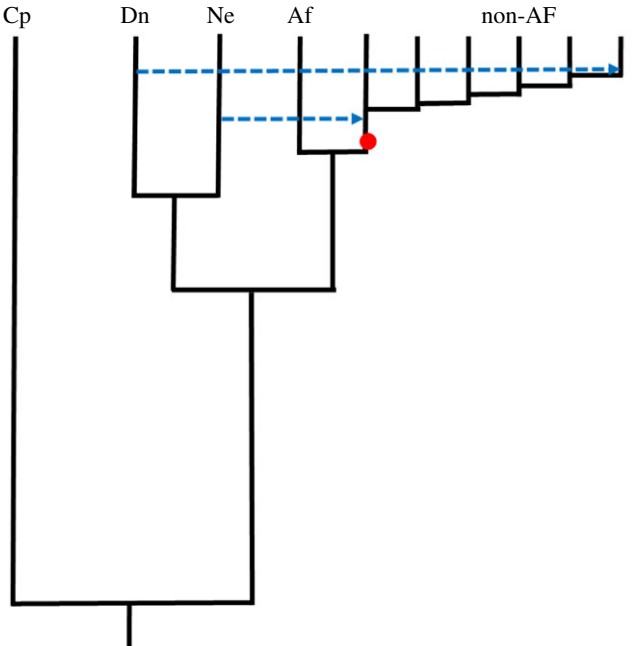

**Figure 9.** Cartoon depicting the structure of coalescent simulations. Taxa simulated were the chimpanzee (Cp), Neanderthal (Ne), Denisova (Dn), African humans (AF) and a series of non-African human populations (non-AF). Dotted blue arrows indicate introgression events and a red dot indicates the out of Africa bottleneck, calibrated to cause a loss of about 25% of heterozygosity. Timings are given in Methods and are not drawn to scale.

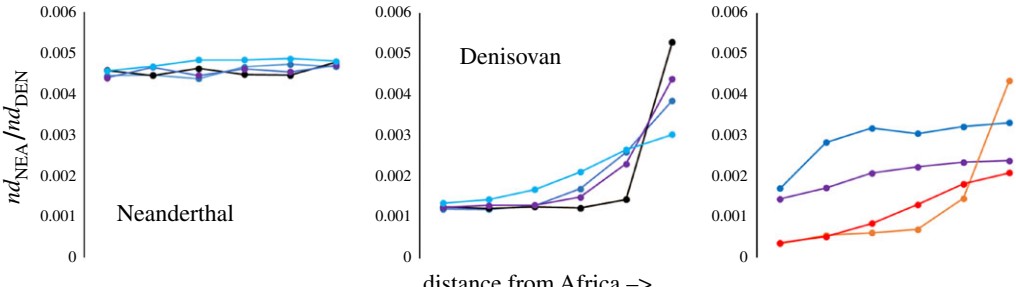

**Figure 10.** Simulated introgression results. Data represent the six non-African populations, arranged from left to right in order of increasing distance from Africa (founding order, figure 9). The left-hand panel shows the effect on $nd_{NEA}$ of varying migration rate between adjacent populations: 1, light blue; 10, dark blue; 20, purple; 50, black. There is no migration in/out of Africa and introgression has occurred between Neanderthals and the population that left Africa prior to further dispersal, while Denisovan introgression occurred into the population furthest from Africa. As seen, the signal is large and does not vary between populations. The middle panel is the same as the left-hand panel, with the same colour-coding, but this time depicts $nd_{DEN}$. As expected, the large Denisovan signal in the population furthest from Africa spreads to the west as migration rates increase. The right-hand panel illustrates the impact of introducing gene flow into and out of Africa for $nd_{NEA}$ (blue, non-African migration rate of 10; purple, non-African migration rate of 50) and $nd_{DEN}$ (orange, non-African migration rate of 10; red, non-African migration rate of 50). Note how migration out of Africa 'dilutes' the Neanderthal signal and has the potential to create similar rising profiles for both archaics.

individuals per generation. Naturally, there are many parameters that can be explored, including the number and strength of founding events, population growth rates, variable migration rates between different populations, the size and timing of introgression and more. Nonetheless, these simulations reveal two important features that appear to be robust. First, in the absence of gene flow in and out of Africa, $nd_{NEA}$ is always constant across all non-African populations (figure 10, left-hand panel). This is in line with theoretical expectations that neutral sequences should not change in average frequency. Second, $nd_{DEN}$ is highest in the population where introgression occurred and decreases east to west at rates that depend on the rate of migration, with low migration rates giving steep, exponential declines and high migration rates

generating shallower, more linear declines (figure 10, middle panel). The shape of the decline is important because, in real data, $nd_{NEA}$ and $nd_{DEN}$ both exhibit approximately linear trends. Finally, adding in migration in and out of Africa acts to 'dilute' the introgression signal towards the west, causing both archaic signals to rise west to east, $nd_{NEA}$ rising because dilution is greatest near Africa while $nd_{DEN}$ rises because introgression occurred in the east. Again, the shape of the trends depends on the level of linking gene flow and the potential exists to create correlated profiles.

In simulations, high migration rates generate approximately linear trends, as seen in real data. However, these migration rates appear high relative to intuitive expectations based on the extent of genetic and phenotypic differentiation of global human populations. As a second test of compatibility, I, therefore, calculated pairwise measures of genetic distance, calculated as the average allele frequency difference (AFD) [40] between each population. Taking the most extreme case of the separation between Europe and East Asia, real data (1000 genomes) reveal a typical value is 0.0135. In simulations, this empirical value is only matched with the lowest migration rate of one per generation, while the highest migration rate gives a value of 0.007, half the empirical value. Essentially identical results are obtained if I use $F_{ST}$ instead of AFD. These findings make intuitive sense in the light of the level of differentiation seen among modern humans, including the dramatic difference in estimated legacy sizes between Papuans and other population groups from Oceania. Thus, there appears to be a contradiction: the observed approximately linear west to east trends in $nd$ and $f_4$, if generated by introgression, would require appreciably higher inter-population mixing than is compatible with observed levels of population differentiation.

# 3. Discussion

Here, I explore the relationship between signals interpreted as evidence of archaic introgression by two archaic hominins, Neanderthals and Denisovans, both with each other and with geography. I find surprising levels of covariation coupled with a joint tendency of both signals to increase from west to east. Moreover, the data downloaded from Mallick et al. [25] reveal a surprising, steeply rising trend in Africa which, if extrapolated, would be consistent with the very high values of Denisovan ancestry reported in Papua New Guinea and Australia [23–25]. A similar 'double trend' is found using a second introgression statistic, $f_4$, in a second large published dataset from Qin & Stoneking [4]. Surprisingly, $f_4$ values where an archaic genome is replaced by that of a great ape are highly correlated with those obtained when the test taxon is Neanderthal, both across human populations and, crucially, across the genome within a particular comparison. Simulations suggest that migration rates among human populations may be too low to generate gradients in legacy through gene flow away from a localized focus of inter-breeding.

All non-Africans are estimated to carry appreciable amounts of Neanderthal DNA, with the lowest value in the Simons Genome Diversity panel being an $nd_{NEA}$ value of 1.5% [25]. Given that Neanderthals appear to have had a restricted distribution focused on western and central Eurasia [41,42], the global non-African legacy is unlikely to arise mainly from local inter-breeding because this would result in substantially lower legacies wherever Neanderthals were not encountered. This truism led to the idea that the primary inter-breeding event occurred in the Levant, soon after humans moved out of Africa, such that all non-Africans carry a legacy regardless of where in the world they ended up. However, this model cannot explain how all the autosomes independently show a pattern where legacy size increases with increasing distance from Africa because neutral drift should cause equal numbers of chromosomes to show positive and negative trends, while positive selection would impact each chromosome to different extents depending on the genes they carry.

Regardless of what happened immediately following the out of Africa event, a Neanderthal legacy that increases west to east is unexpected. Coalescent simulations confirm that legacy size is always extremely stable across daughter populations following inter-breeding. The primary exception to this is when a legacy is diluted by mixing with a population with a lower/zero legacy, for example, via gene flow out of Africa. However, legacy size should never increase unless further inter-breeding occurs with archaics. Consequently, the largest inferred legacies in extant human populations are expected always to involve populations whose ancestors received the largest archaic inputs. For both Neanderthals and Denisovans, but particularly Neanderthals, this means inter-breeding was greatest with the ancestors of people who now live in parts of the world that lie furthest from Africa such as Northeast Siberia, Japan, Southeast Asia and Oceania. These regions lie well outside the range currently understood to have been occupied by Neanderthals [41,42].

Of course, a population's current location may lie a long way from where any inter-breeding occurred, yet a plausible scenario remains difficult to find. If inter-breeding occurred, it must have taken place before Neanderthals went extinct and probably in western Eurasia where archaeology indicates most Neanderthals lived [41,42]. The resulting legacy would have been carried further away from Africa as people moved, but it would have stayed at the same frequency both in populations that stayed near where the inter-breeding happened and in descendent populations that migrated, giving an overall flat distribution. The data could be explained by more complicated scenarios. For example, if two waves of humans spread across the world, the first carrying a much larger legacy than the second, inter-breeding between the two could generate a west to east trend. Unfortunately, such models raise other issues, such as how they fit with the strong, apparently uniform linear decline in heterozygosity with distance from Africa [37], how the two waves were formed and why mitochondrial DNA variants do not appear to betray their existence.

Recent work suggests that archaic legacies in Africa can be non-zero [28], but the strongly correlated $nd_{NEA}$ and $nd_{DEN}$ values among some African populations still sit somewhat uncomfortably with current knowledge. First, it appears to require a second, entirely independent instance of 'coordinated' introgression in a part of the world where neither archaic appears to have lived. Second, at its peak, $nd_{DEN}$ in Africa is higher than anywhere in Eurasia, making it difficult to dismiss this pattern as genetic leakage via back-migrations of non-Africans back into Africa. Moreover, the slope in Africa is radically different from the trend seen across the whole of Eurasia, being close to an inverse trend, with $nd_{DEN}$ being approximately 2.5× higher than $nd_{NEA}$ inside Africa, and $nd_{NEA}$ being approximately 2.5-fold higher than $nd_{DEN}$ outside Africa. It remains unclear whether this possible relationship is coincidence, just as it remains unclear whether the high values of $nd_{DEN}$ found in Oceania/Australia reflect an extension of the African trend. In this context, it is interesting that $f_4$ statistics in a second, large dataset also indicate two distinct, contrasting trends.

One issue that should not be ignored is the performance of the statistics used to infer introgression. The $nd$ statistic attempts to separate signals due to the two different archaics. However, $nd$ requires bases to be classified as ancestral or derived within each archaic, despite being determined in, in most cases, a single reference genome. Consequently, there will be many instances where bases are mis-classified. Wherever a variant that is found in both archaic populations but is reported in the reference sequence of only one, the result will signal 'leakage': introgressed sequences from one species are able to contribute to an apparent signal of the other. The potential exists, therefore, for correlations between $nd_{NEA}$ and $nd_{DEN}$ to arise through incomplete signal separation. In practice, simulated data show that such correlations, though present, appear too weak to drive the observed correlations. This is also confirmed by real data by the way that different populations with very similar $nd$ values for one archaic can have radically different values for the other archaic, the most obvious example being Papuans and Australians who have similar $nd_{NEA}$ values to many East Asian populations but $nd_{DEN}$ values that are around fivefold larger. Nonetheless, simulations do suggest that correlations are possible if there is an appreciable gene flow out of Africa that acts to dilute Neanderthal legacies in the west, though more work is needed to assess whether actual levels could have been sufficient.

A more striking issue is revealed in the $f_4$ statistic. Here, values that should capture introgression by Neanderthals and Denisovans in the same set of human populations are highly correlated. The strength of the correlation appears intuitively at odds with widespread observations that the sizes of the two legacies vary considerably in their relative proportions in different human populations. Moreover, the strong correlations between paired $f_4$ values where one archaic is substituted by the other extends to comparisons where one archaic is substituted by a great ape. Since introgression of great apes into humans is biologically implausible, this observation suggests that $f_4$ is, at least in some instances, capturing a signal that is not due to introgression. Crucially, $f_4(ESN, Neanderthal; GBR, X)$ correlates with $f_4(ESN, great ape; GBR, X)$ when $X$ is one of a range of different populations in the 1000 genomes dataset. Such a correlation makes it difficult to argue convincingly that significantly non-zero $f_4(ESN, Neanderthal; GBR, X)$ is due to Neanderthal introgression while significantly non-zero $f_4(ESN, great ape; GBR, X)$ is an artefact. The intimate relationship between $f_4$ values featuring archaics and their equivalent values featuring great apes is further emphasized by the way that, for any given population $X$, the paired values are highly correlated across the genome. To me, this appears to preclude introgression as the primary driver. Instead, a more parsimonious explanation is that the outgroups, whether archaic hominins or a great ape, are both acting as markers of the ancestral state and this is why both yield similar signals both, across different human population combinations and, within any one comparison, across the genome.

Positive signals of introgression in biologically implausible scenarios featuring great apes are not the only problem to face the currently accepted models of human–archaic inter-breeding. I have recently shown that positive $D$(African, non-African, Neanderthal, chimpanzee) statistics are driven not by heterozygous sites in *non-Africans*, as expected of rare introgressed fragments, but by heterozygous sites in *Africans* [30]. This analysis suggests that introgression statistics are driven more by a higher mutation rate in Africa, driving faster divergence from the ancestral state rather than by introgressed fragments outside Africa acting to reduce divergence. Here, the signal captured by $D$ stems from mutation rate variation within humans and, as a result, the archaic genome acts merely as a marker of the ancestral state. This result is consistent with results presents here, where the two archaics can be substituted by great apes without abolishing the signal: essentially the outgroup does not matter as long as it is close enough to humans to provide a marker for the ancestral state.

As discussed in the introduction, introgression statistics capture asymmetrical branch lengths and can arise either by introgression acting to shorten a branch or through a higher mutation rate acting to lengthen a branch. Of these, only the former has been given any credence, with mutation rate variation generally being explicitly assumed not to exist. In fact, mutation rate differences between human populations do exist. Although these are small when estimated over each population's entire history [25], when analyses focus on variants that arose after the out of Africa event(s), much larger differences are seen, both for different mutation types [34] and as a higher overall rate in Africa [31]. I have previously speculated that a mutation rate variation might be driven by HI, the idea that heterozygosity modulates mutation rate such that the large loss of heterozygosity suffered by non-Africans during the out of Africa bottleneck caused a parallel slowdown in mutation rate [31,32,43]. The current results lend further support to this alternative model. Heterozygosity and other measures of genetic variability all exhibit a striking negative correlation with distance from Africa [37,44,45]. If heterozygosity and mutation rate are indeed correlated, the well-documented trend in heterozygosity would generate a parallel trend, with genetic distance to our nearest relatives decreasing with increasing distance from Africa, as observed. This model may be speculative and certainly requires more testing, but it does help to explain both why all non-Africans carry signals linked to both of the archaics (and great apes!) and why the estimated peaks in legacy size tend to lie, not in one cohesive region that archaics may reasonably have occupied, but instead are spread across a vast area whose only common property is that they lie furthest from Africa.

An obvious question that arises concerns the origin of the very high $nd_{DEN}$ values in Oceania/Australia. There appear to be three possibilities. First, these may reflect genuine legacies due to inter-breeding. This may well be the case, though, like the rising $nd_{NEA}$ values across Eurasia, it remains difficult to envisage where and when the main inter-breeding event occurred. Second, under the HI hypothesis, a further bottleneck between East Asia and Oceania could have caused a further drop in mutation rate. This possibility seems unlikely to generate the required effect size and fits poorly with the existence of two clearly distinct trends for both $nd$ and $f_4$, with values that imply radically different ratios of Neanderthal to Denisovan fraction. Third, the very high $nd_{DEN}$ values coupled with some of the highest $nd_{NEA}$ values anywhere in the world place Papuans and Australians at the end of an apparent steep trend in Africa. This raises the possibility of two out of Africa events, one leading to the peopling of parts of Oceania and Australia and a later event that led to the peopling of the remainder of the non-African world. Such a scenario has received recent support [26]. An earlier event would give more time for the impact of mutation rate changes to unwind, leading to higher $nd$ values, while a different progenitor African lineage that was relatively closer to Denisovans might account for the different slope. The mutation slowdown model also offers a solution to the paradox that relatively smooth geographical trends suggest high levels of gene flow between neighbouring populations yet Papuans live near other populations with substantially lower Denisovan legacies, implying very low levels of mixing. The mutation slowdown model breaks this paradox by removing the need for extensive mixing: the geographical trend is generated instead by the progressive loss of diversity as the world was colonized. With low levels of mixing, if there were indeed two waves, these could have remained largely distinct. The mutation slowdown model thus fits well with several key features of the real data, but it remains speculative and more work is needed to understand the dynamics of how mutation slowdown might work in practice.

Finally, I often meet the argument that the identification of introgressed haplotypes offers the 'gold standard' method for quantifying introgression. However, the underlying signal is the same and requires largely the same if not more assumptions to be quantified. Thus, the most clearly described method of inference I could find is in Skov *et al.* [15]. Here, they use criteria that are closely allied to the rationale that underpins $nd$ (a focus on derived archaic variants that are present outside Africa

and absent from a reference panel of sub-Saharan Africans) to identify candidate introgressed bases and then use a Markov model to search for significant clustering. As with $D$, $nd$ and $f_4$, this method should work very well if, but only if, mutations occur at a constant rate, are almost never recurrent and occur independently. The current study, by challenging the assumption that individual introgressed bases can be identified with confidence, must also raise questions about what clusters of such bases represent.

In conclusion, although the distribution of inferred archaic legacies has not, until now, been seen as particularly problematic, I show that they are difficult to explain by the classic inter-breeding model whereby gene flow from Neanderthals entered non-Africans in the Levant, soon after the out of Africa event, and from Denisovans sometime later in East Asia. The key issues are the consistency of signal across all chromosomes, which indicates that global variation in legacy size is not generated by either drift or selection, and the levels of gene flow linking human populations, which seem too low to generate the observed gradients in legacy. Together, these issues make an increasing west to east trend difficult to reconcile with inter-breeding that occurred mainly in west Eurasia. However, this trend fits well with a model based on heterozygosity acting to modulate mutation rate and thereby the extent that different populations have diverged from our common ancestor. I hope this analysis causes future work to revisit the twin untested assumptions that underpin all estimates of archaic legacy, namely that mutation rate does not vary between populations and that back-mutations are sufficiently rare that they can be safely ignored.

# 4. Methods

## 4.1. Data

Data were downloaded from Phase 3 of the 1000 genomes project [46] as composite vcf files from (ftp://ftp.1000genomes.ebi.ac.uk/vol1/ftp/release/20130502/). These comprise low coverage genome sequences for 2504 individuals drawn from 26 modern human populations spread across five geographical regions: Europe (GBR, FIN, CEU, IBS, TSI); East Asia (CHB, CHS, CDX, KHV, JPT); Central Southern Asia (GIH, STU, ITU, PJL, BEB); Africa (LWK, ESN, MSL, YRI, GWD, YRI, ASW, ACB) and the Americas (MXL, CLM, PUR, PEL). Populations are listed in the order they appear in the analysis (GBR = 1, FIN = 2 . . . PEL = 26). Individual chromosome vcf files for the Altai Neanderthal genome were downloaded from http://cdna.eva.mpg.de/neandertal/altai/AltaiNeandertal/VCF/. Individual chromosome vcf files for the Denisovan genome were downloaded from http://cdna.eva.mpg.de/denisova/VCF/human/. For analyses presented here, I focused only on homozygote archaic bases, accepting only those with 10 or more reads, fewer than 250 reads and where more than 80% were of one particular base. This approach sacrifices modest numbers of (usually uninformative) heterozygous sites but benefit from avoiding ambiguities caused by coercing low counts into genotypes.

## 4.2. Data analysis

Analysis of the 1000 genomes data and archaic genomes was conducted using custom scripts written in C++ and are minor variants of the code given in Amos [30]. The actual code used is available in electronic supplementary material. The statistics $nd_{10}$ and $nd_{01}$ attempt to capture the probability that a base in a non-African population is of Neanderthal or Denisovan ancestry, respectively. Given that the '01' and '10' subscripts are often difficult to read and easy to confuse (and limited to two different archaics!) I prefer to use the nomenclature $nd_{NEA}$ and $nd_{DEN}$. Each measure is calculated probabilistically as the chance that a randomly selected base is, for $nd_{NEA}$, derived in Neanderthal, ancestral in the Denisovan, chimpanzee and a panel of sub-Saharan Africans and derived in the population of interest. To convert this to a proportion of the genome, French are assumed to carry 2% Neanderthal DNA, Papuans are assumed to carry 5% Denisovan DNA and these 'fixed' values then used as references. For example, an $nd_{NEA}$ value that is 1.5× the French $nd_{NEA}$ value would be converted to a genomic fraction of 3%. This method is challenging in practice because it requires every base in the genomes of all taxa to be scored according to quality and type. As a more workable compromise, and because I use $nd$ more as a comparative measure, I used as the divisor the total number of biallelic and hence informative sites in humans. Thus, $nd = \sum d_i / N$, where $N$ is the total number of biallelic sites in humans where base calls have been made for all taxa and $d_i$ is the frequency of the $i$th derived allele in the population of interest, a derived allele being defined as an allele that is present in the focal archaic and not present in either the other archaic or the chimpanzee or a panel of sub-Saharan Africans.

## 4.3. Simulations

Simulated data were generated using code originally written for Hudson's coalescent simulation program ms [39]. The following is an example of the code used:

```
./ms 741 1000 -t 200 -I 10 1 20 20 100 100 100 100 100 100 100 -r 100 25000 -m 5 6 1 -m 6 5 1 -m 6 7 1 -m
7 6 1 -m 7 8 1 -m 8 7 1 -m 8 9 1 -m 9 8 1 -m 9 10 1 -m 10 9 1 -es 0.04 10 0.97 -ej 0.0401 11 3 -es 0.06 5 0.97
-ej 0.0601 12 2 -ej 0.042 10 9 -ej 0.044 9 8 -ej 0.046 8 7 -ej 0.048 7 6 -ej 0.05 6 5 -ej 0.07 5 4 -en 0.0685 5 0.007 -ej
0.4 3 2 -ej 0.5 4 2 -ej 6 2 1
```

I assume a haploid population size of 10 000, a mutation rate of $10^{-8}$ and set $\theta$ to 200, such that each of 1000 fragments is 500 kb long. Trials with much larger numbers of shorter fragments and no recombination gave essentially identical results, but to allow an impact, if any, of recombination, the results presented here are for the longer fragments. I assume a hominin–chimpanzee split occurs at 6 000 000 years ago, the archaic–human split is set at 500 000 years ago (generation length = 25 years), Neanderthals and Denisovans split at 450 000 years ago and the out of Africa event is placed at 70 000 years ago. Immediately after the 'out of Africa' event a population bottleneck is set to cause a realistic average loss of 25% of initial heterozygosity. At 60 000 years ago, a 3.4% pulse of Neanderthal DNA is injected into the non-African lineage. Without Denisovan introgression, this generates a realistic D of approximately 4%. Starting at 50 000 years ago, the non-African lineage splits to form a daughter population that in turn splits 2000 years later. Such splitting is repeated at the same interval until six non-African populations have been generated. Sequentially generated non-African populations are linked by bidirectional migration. At 40 000 years ago, the last human population to be formed receives a 3.4% pulse of Denisovan DNA. Migration rates were varied between one and 100 individuals per generation and I also explored adding a modest extra bottleneck after each non-African population split, but the impact was negligible so this step was omitted from the final results.

Data accessibility. Data for analysis of the 1000 genomes project data are available as a C++ script (electronic supplementary material) and data files (Dryad Digital Repository: Neanderthal and chimpanzee alignment files are available at https://doi.org/10.5061/dryad.2fqz612kn as part of a prior publication [47], the additional Denisovan files required for the current publication are available at the pre-publication link https://doi.org/10.5061/dryad.0cfxpnw01). Data from the Simons Genome Diversity Project were downloaded from https://www.ncbi.nlm.nih.gov/pmc/articles/PMC5161557/bin/NIHMS798259-supplement-supp_datatable1.xlsx, which is freely available, confirmed by senior author David Reich.

Competing interests. This publication is a continuation of work that formed the basis of a public challenge presented in the form of a cash bet. The public challenge was a five-figure cash bet offered from 2017 following the posting of a related preprint (https://www.biorxiv.org/content/early/2017/05/03/133306) to encourage the community to engage with the research questions posed. At the time of acceptance of the paper, details of the challenge were available at my website https://www.zoo.cam.ac.uk/directory/william-amos and https://www.researchgate.net/project/Neanderthal-introgression-a-case-of-smoke-and-mirrors. However, the current paper has no overlap with this earlier preprint and the challenge itself expired in June 2019.

Funding. This work was not funded.

Acknowledgements. I am indebted to Simon Martin, Tom van de Valk, Rob Foley and Pavel Flegontov for useful discussions.

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
