## [Peer Review File · Royal Society Open Science]

Review History

RSOS-201229.R0 (Original submission)

Review form: Reviewer 1

Is the manuscript scientifically sound in its present form?

No

Are the interpretations and conclusions justified by the results?

No

Is the language acceptable?

Yes

Do you have any ethical concerns with this paper?

No

Have you any concerns about statistical analyses in this paper?

No

Recommendation?

Major revision is needed (please make suggestions in comments)

Comments to the Author(s)

I In his manuscript "Correlated and geographically predictable Neanderthal and Denisovan legacies are difficult to reconcile with a simple model based on inter-breeding" the author cautions for the presence of artefacts or issues previously unaccounted for when describing the otherwise commonly accepted interaction between AMH and Neanderthal or Denisova.

I am not a native speaker, but I have some practical concerns on the style, since the manuscript is written almost as an essay rather than as a scientific article, and I think it would benefit from a small restructure, to highlight better what are the 1) Problem 1; 2) Problem 2; 3) Problem 3 etc.. and to explain in a schematic way how the analyses proposed here address those problems and perhaps elucidate a more plausible alternative model which should be adopted instead. For example it is not entirely clear whether the author is arguing a) against the idea of Neanderthal and Denisova archaic introgressions OoA or b) the current space/time models we have to reconstruct those events or c) the alleged poor performance of nd10NEA and nd10DEN to separate the two archaic contributions or d) a mix of the previous three points.

From my understanding of the author's concerns, essentially the two main points are: 1) Allele sharing stats calculated using nd10NEA and nd10DEN correlate more than expected by chance (it would be nice to see some correlation tests rather than just scatter plots) and 2) There is a geographic pattern (higher archaic legacy with higher distance OoA) which is not expected under either Neanderthal or Denisova introgression models and 3) additionally, in the discussion, the author advocates for putative role of reduced mutation rate OoA.

My objections to the points above are:

1) Given the small number of archaic samples available, the nd10 approach is by no means "water-tight", since variants present in the available Denisova but not in the available Neanderthal genomes (and vice versa) are not granted to be "Denisova-specific" or "Neanderthal specific", and could instead fall on the shared Neanderthal-Denisova stem. Under this assumption (the author could prove me wrong with additional tests), the correlation observed between nd10NEA and nd10DEN could simply be the reflection of the contribution to either statistics of the shared Nea-Deni alleles. That said, if the author has concerns with this particular statistic (nd10), I don't see why he couldn't simply adopt one of the many others available (including those that deal with haplotypes, rather than just single sites) to test the otherwise accepted scenario;

2) Geographic pattern: given that both in real data and in the simulations provided here one of the populations further from Africa (Papuan and to a lesser extent East Asians) have a boost of Denisova introgression, in light of the effect explained in the previous point it is not surprising that also the nd10NEA experience such a correlation. Indeed, while the amount of true Neanderthal component may remain the same across all Eurasia, the latent Denisova component that diffuses westward from Papua/East Asia through isolation by distance can mimic a higher Neanderthal contribution.

3) I didn't entirely get the role of reduced mutation rate in the overall criticism offered by the author, however I think that invoking longer generation times OoA as an alternative/concurrent explanation may result in a similar effect, and perhaps into a more readily acceptable one by the non expert readership.

Minor point:

please include color legends in the figures for ease of reading and if possible also pearson r and p-values when you refer to correlations.

Review form: Reviewer 2

Is the manuscript scientifically sound in its present form?

Yes

Are the interpretations and conclusions justified by the results?

Yes

Is the language acceptable?

Yes

Do you have any ethical concerns with this paper?

No

Have you any concerns about statistical analyses in this paper?

No

Recommendation?

Accept with minor revision (please list in comments)

Comments to the Author(s)

This paper concerns geographic trends of archaic (Neanderthal and Denisovan) legacies in modern human genomes, analyzed based on data from several sources and simple models of migration and introgression, as well as computer simulations.

The author finds a number of inconsistencies of data compared to theoretical models. One of them is the complete reversal of correlations of $nd10(NEA)$ vs. $nd10(DEN)$ (Figure 1), depending on whether they are conditional on absence of archaic admixture in subsaharan Africa (predominant hypothesis) or they are unconditional. On the other hand, a $nd10(NEA)$ vs. $nd10(DEN)$ correlation among Africans follows a complicated pattern (Figure 2) although it is not clear how reverse-migration flow might contribute to it. Influence of selection less likely, since the principal trends are shared in all chromosomes. Another interesting result is summarized in Fig.5 and it shows that the residuals of Neanderthal $nd10$ statistics corrected for land-distance from Africa, follow an East-West increase of Neanderthal legacy, which is partly inconsistent with the geographic range of Neanderthals (highest residuals in China).

A Discussion follows, which finds these findings in contradiction to any simple models of migration-related introgressions. Data might be found consistent with a model assuming that mutation rate is positively correlated with heterozygosity, while the latter decreases with the distance from Africa. However this relationship, once proposed by Amos, is by his own admission, not commonly accepted. One of the attractive features of the paper is a simulation study, which puts together the major tenets (however, please see further on). The most general conclusion of the work is that if the introgressions are real (the Neanderthal one seemingly founded on more extensive evidence), then further effort is needed to understand their nature.

I found one aspect of the paper which needs some improvement. The Discussion of simulation results might be made clearer, for example by making direct references to color-coded lines

plotted in Figure 7 (referring to various cases which correspond or do not correspond to data). This reader seems somewhat lost without this and it can be fixed with relatively simple text alterations.

Figure 5: Color scale seems more like yellow - brown - purple -violet, than what is claimed in the legend (yellow - brown - mauve - blue). I would use a brighter color scale, if possible.

Review form: Reviewer 3

Is the manuscript scientifically sound in its present form?

Yes

Are the interpretations and conclusions justified by the results?

Yes

Is the language acceptable?

Yes

Do you have any ethical concerns with this paper?

No

Have you any concerns about statistical analyses in this paper?

No

Recommendation?

Major revision is needed (please make suggestions in comments)

Comments to the Author(s)

Comments are included in the attached pdf (Appendix A).

Decision letter (RSOS-201229.R0)

Dear Dr Amos

The Editors assigned to your paper RSOS-201229 "Correlated and geographically predictable Neanderthal and Denisovan legacies are difficult to reconcile with a simple model based on inter-breeding." have now received comments from reviewers and would like you to revise the paper in accordance with the reviewer comments and any comments from the Editors. Please note this decision does not guarantee eventual acceptance.

Please submit your revised manuscript and required files (see below) no later than 21 days from today's (ie 02-Mar-2021) date. Note: the ScholarOne system will 'lock' if submission of the revision is attempted 21 or more days after the deadline. If you do not think you will be able to meet this deadline please contact the editorial office immediately.

on behalf of Professor Matthew Collins (Associate Editor) and Kevin Padian (Subject Editor)
openscience@royalsociety.org

Subject Editor Comments to Author (Professor Kevin Padian):

Thank you for your submission. As you will see, the reviewers had some concerns that likely will need to be addressed with a bit more analysis. As such, if you need more time than the three weeks we customarily allot to major revisions, please let the editorial office know. Your resubmission may be returned to some reviewers for further assessment, so please take care to respond to their concerns. Best wishes.

Associate Editor Comments to Author (Professor Matthew Collins):
Comments to the Author:
Dear Prof Amos

I must apologize for the time it has taken (7 months) to review this MS which stemmed from challenge of finding reviewers for it, and then a number of reviewers who failed to provide reviews. Indeed your paper has been the most challenging to review of any that I have edited.

Overall I think that the Abstract is slightly overselling the paper. Otherwise I see the comments as addressable by a minor review. The most significant issues to address are those raised by Reviewer #1 and regard the use of nd10.

Reviewer #1

In his manuscript "Correlated and geographically predictable Neanderthal and Denisovan legacies are difficult to reconcile with a simple model based on inter-breeding" the author cautions for the presence of artefacts or issues previously unaccounted for when describing the otherwise commonly accepted interaction between AMH and Neanderthal or Denisova.

I am not a native speaker, but I have some practical concerns on the style, since the manuscript is written almost as an essay rather than as a scientific article, and I think it would benefit from a small restructuring, to highlight better what are the 1) Problem 1; 2) Problem 2; 3) Problem 3 etc.. and to explain in a schematic way how the analyses proposed here address those problems and perhaps elucidate a more plausible alternative model which should be adopted instead.

For example it is not entirely clear whether the author is arguing

- a) against the idea of Neanderthal and Denisova archaic introgressions OoA or
- b) the current space/time models we have to reconstruct those events or
- c) the alleged poor performance of nd10NEA and nd10DEN to separate the two archaic contributions or
- d) a mix of the previous three points.

From my understanding of the author's concerns, essentially the two main points are:

- 1) Allele sharing stats calculated using nd10NEA and nd10DEN correlate more than would be expected by chance (it would be nice to see some correlation tests rather than just scatter plots) and
- 2) There is a geographic pattern (higher archaic legacy with higher distance OoA) which is not expected under either Neanderthal or Denisova introgression models and
- 3) additionally, in the discussion, the author advocates for putative role of reduced mutation rate OoA.

My objections to the points above are:

1) Given the small number of archaic samples available, the nd10 approach is by no means "water-tight", since variants present in the available Denisova but not in the available Neanderthal genomes (and vice versa) are not granted to be "Denisova-specific" or "Neanderthal specific", and could instead fall on the shared Neanderthal-Denisova stem. Under this assumption (the author could prove me wrong with additional tests), the correlation observed between nd10NEA and nd10DEN could simply be the reflection of the contribution to either statistics of the shared Nea-Deni alleles. That said, if the author has concerns with this particular statistic (nd10), I don't see why he couldn't simply adopt one of the many others available (including those that deal with haplotypes, rather than just single sites) to test the otherwise accepted scenario;

2) Geographic pattern: given that both in real data and in the simulations provided here one of the populations further from Africa (Papua and to a lesser extent East Asians) have a boost of Denisova introgression, in light of the effect explained in the previous point it is not surprising that also the nd10NEA experience such a correlation. Indeed, while the amount of true Neanderthal component may remain the same across all Eurasia, the latent Denisova component that diffuses westward from Papua/East Asia through isolation by distance can mimic a higher Neanderthal contribution.

3) I didn't entirely get the role of reduced mutation rate in the overall criticism offered by the author, however I think that invoking longer generation times OoA as an alternative/concurrent explanation may result in a similar effect, and perhaps into a more readily acceptable one by the non expert readership.

Minor point:

Please include color legends in the figures for ease of reading and if possible also pearson r and p-values when you refer to correlations.

Review #2

This paper concerns geographic trends of archaic (Neanderthal and Denisovan) legacies in modern human genomes, analyzed based on data from several sources and simple models of migration and introgression, as well as computer simulations.

The author finds a number of inconsistencies of data compared to theoretical models. One of them is the complete reversal of correlations of nd10(NEA) vs. nd10(DEN) (Figure 1), depending on whether they are conditional on absence of archaic admixture in subsaharan Africa (predominant hypothesis) or they are unconditional. On the other hand, a nd10(NEA) vs. nd10(DEN) correlation among Africans follows a complicated pattern (Figure 2) although it is not clear how reverse-migration flow might contribute to it. Influence of selection is less likely, since the principal trends are shared in all chromosomes. Another interesting result is summarized in Fig.5 and it shows that the residuals of Neanderthal nd10 statistics corrected for land-distance from Africa, follow an East-West increase of Neanderthal legacy, which is partly inconsistent with the geographic range of Neanderthals (highest residuals in China).

A Discussion follows, which finds these findings in contradiction to any simple models of migration-related introgressions. Data might be found consistent with a model assuming that mutation rate is positively correlated with heterozygosity, while the latter decreases with the distance from Africa. However this relationship, once proposed by Amos, is by his own admission, not commonly accepted. One of the attractive features of the paper is a simulation study, which puts together the major tenets (however, please see further). The most general conclusion of the work is that if the introgressions are real (the Neanderthal one seemingly founded on more extensive evidence), then further effort is needed to understand their nature.

I found one aspect of the paper which needs some improvement. The Discussion of simulation results might be made clearer, for example by making direct references to color-coded lines plotted in Figure 7 (referring to various cases which correspond or do not correspond to data). This reader seems somewhat lost without this and it can be fixed with relatively simple text alterations.

Figure 5: Color scale seems more like yellow - brown - purple -violet, than what is claimed in the legend (yellow - brown - mauve - blue). I would use a brighter color scale, if possible.

Reviewer #3:

Overall this paper utilized a five-populations allele sharing statistic to characterize the possible correlations

- i) between Denisovan and Neandertal introgression in worldwide populations,
- ii) between archaic introgression and geography.

The author tested these correlations via simulated data, publicly available 1000Genomes and SGDP datasets. The results presented in this paper are intriguing, though the results can be presented in a clearer way and interpreted more carefully (see comments below).

Major Comments:

1. The major conclusion of this paper relies on interpreting the nd10 results. The author explains nd10 in the first paragraph of page 6. By introducing five taxa in the paragraph (in contrast to

four taxa in D statistics), I think it adds difficulty for readers to understand. It would be good to present nd_{10} statistics in a simply one-line formula to make it easy to understand what nd_{10} has been calculated.

2. Line 159, is figure 3 a zoom-in plot of figure 1? I am very confused because in the legend it says these are nd_{10} values from Mallick et al 2020. It should be better explained.

3. The author has mentioned repeatedly in the paper that the archaic legacy size is stable and described nd_{10} calculation for each chromosome in line 134-146. However there is zero plot on per-chromosome nd_{10} results in this paper...

e.g. in the Abstract: "Simulations confirm that, once created, legacy size is extremely stable: it may reduce through admixture with lower legacy populations but cannot increase detectably through neutral drift".

In Discussion: "The key issues are the consistency of signal across all chromosomes, which indicates that global variation in legacy size is not generated by either drift or selection".

4. In general I found the statements on mutation rate to be confusing. They read clearly in the introduction which serves as background but fit less well in the discussion. If the author insists on keeping these statements, please check their consistency between sections. Also it is important to distinguish between results, interpretations and speculations in the Discussion. Some sentences combine interpretation and speculation, and it would be useful to wordsmith these to discriminate between results and speculation.

Minor Comments:

1. Line 102-104, to XXBAA, XXABA, XXBBA more understandable, it would be good to have a half sentence or a sentence to explain, e.g. XXBAA representing Denisovan and Chimpanzee alleles agree

2. Line 111-113, this sentence corresponds well to the results presented in figure 1. But how nd_{10} was calculated for the rest of figures still needs to be explained - with or without conditioning on the African state and the use of transitions or transversions?

3. Line 189, " nd_{10NEA} is always constant across all non-African populations". I do not see the result for this. The legend of figure 7 says all values line between 0.0155 and 0.0165. It would be good to have a panel in parallel in figure 7 showing nd_{10NEA} values.

4. line 200, could the result be replicated with F_{st} , instead of AFD? 5. line 500, 'distance' typo

6. Figure 7, better to illustrate x-axis though there are some texts in the legend.

Reviewer comments to Author:

Reviewer: 1

Comments to the Author(s)

I In his manuscript "Correlated and geographically predictable Neanderthal and Denisovan legacies are difficult to reconcile with a simple model based on inter-breeding" the author cautions for the presence of artefacts or issues previously unaccounted for when describing the otherwise commonly accepted interaction between AMH and Neanderthal or Denisova.

I am not a native speaker, but I have some practical concerns on the style, since the manuscript is written almost as an essay rather than as a scientific article, and I think it would benefit from a small restructure, to highlight better what are the 1) Problem 1; 2) Problem 2; 3) Problem 3 etc.. and to explain in a schematic way how the analyses proposed here address those problems and perhaps elucidate a more plausible alternative model which should be adopted instead. For example it is not entirely clear whether the author is arguing a) against the idea of Neanderthal and Denisova archaic introgressions OoA or b) the current space/time models we have to reconstruct those events or c) the alleged poor performance of nd_{10NEA} and nd_{10DEN} to separate the two archaic contributions or d) a mix of the previous three points.

From my understanding of the author's concerns, essentially the two main points are: 1) Allele sharing stats calculated using nd10NEA and nd10DEN correlate more than expected by chance (it would be nice to see some correlation tests rather than just scatter plots) and 2) There is a geographic pattern (higher archaic legacy with higher distance OoA) which is not expected under either Neanderthal or Denisova introgression models and 3) additionally, in the discussion, the author advocates for putative role of reduced mutation rate OoA.

My objections to the points above are:

1) Given the small number of archaic samples available, the nd10 approach is by no means "water-tight", since variants present in the available Denisova but not in the available Neanderthal genomes (and vice versa) are not granted to be "Denisova-specific" or "Neanderthal specific", and could instead fall on the shared Neanderthal-Denisova stem. Under this assumption (the author could prove me wrong with additional tests), the correlation observed between nd10NEA and nd10DEN could simply be the reflection of the contribution to either statistics of the shared Nea-Deni alleles. That said, if the author has concerns with this particular statistic (nd10), I don't see why he couldn't simply adopt one of the many others available (including those that deal with haplotypes, rather than just single sites) to test the otherwise accepted scenario;

2) Geographic pattern: given that both in real data and in the simulations provided here one of the populations further from Africa (Papuan and to a lesser extent East Asians) have a boost of Denisova introgression, in light of the effect explained in the previous point it is not surprising that also the nd10NEA experience such a correlation. Indeed, while the amount of true Neanderthal component may remain the same across all Eurasia, the latent Denisova component that diffuses westward from Papua/East Asia through isolation by distance can mimic a higher Neanderthal contribution.

3) I didn't entirely get the role of reduced mutation rate in the overall criticism offered by the author, however I think that invoking longer generation times OoA as an alternative/concurrent explanation may result in a similar effect, and perhaps into a more readily acceptable one by the non expert readership.

Minor point:

please include color legends in the figures for ease of reading and if possible also pearson r and p-values when you refer to correlations.

Reviewer: 2

Comments to the Author(s)

This paper concerns geographic trends of archaic (Neanderthal and Denisovan) legacies in modern human genomes, analyzed based on data from several sources and simple models of migration and introgression, as well as computer simulations.

The author finds a number of inconsistencies of data compared to theoretical models. One of them is the complete reversal of correlations of nd10(NEA) vs. nd10(DEN) (Figure 1), depending on whether they are conditional on absence of archaic admixture in subsaharan Africa (predominant hypothesis) or they are unconditional. On the other hand, a nd10(NEA) vs. nd10(DEN) correlation among Africans follows a complicated pattern (Figure 2) although it is not clear how reverse-migration flow might contribute to it. Influence of selection less likely, since the principal trends are shared in all chromosomes. Another interesting result is summarized in Fig.5 and it shows that the residuals of Neanderthal nd10 statistics corrected for land-distance from

Africa, follow an East-West increase of Neanderthal legacy, which is partly inconsistent with the geographic range of Neanderthals (highest residuals in China).

A Discussion follows, which finds these findings in contradiction to any simple models of migration-related introgressions. Data might be found consistent with a model assuming that mutation rate is positively correlated with heterozygosity, while the latter decreases with the distance from Africa. However this relationship, once proposed by Amos, is by his own admission, not commonly accepted. One of the attractive features of the paper is a simulation study, which puts together the major tenets (however, please see further on). The most general conclusion of the work is that if the introgressions are real (the Neanderthal one seemingly founded on more extensive evidence), then further effort is needed to understand their nature.

I found one aspect of the paper which needs some improvement. The Discussion of simulation results might be made clearer, for example by making direct references to color-coded lines plotted in Figure 7 (referring to various cases which correspond or do not correspond to data). This reader seems somewhat lost without this and it can be fixed with relatively simple text alterations.

Figure 5: Color scale seems more like yellow - brown - purple -violet, than what is claimed in the legend (yellow - brown - mauve - blue). I would use a brighter color scale, if possible.

Reviewer: 3

Comments to the Author(s)
comments are included in the attached pdf.

===PREPARING YOUR MANUSCRIPT===

Your revised paper should include the changes requested by the referees and Editors of your manuscript. You should provide two versions of this manuscript and both versions must be provided in an editable format:
one version identifying all the changes that have been made (for instance, in coloured highlight, in bold text, or tracked changes);
a 'clean' version of the new manuscript that incorporates the changes made, but does not highlight them. This version will be used for typesetting if your manuscript is accepted.
Please ensure that any equations included in the paper are editable text and not embedded images.

If you have been asked to revise the written English in your submission as a condition of publication, you must do so, and you are expected to provide evidence that you have received language editing support. The journal would prefer that you use a professional language editing service and provide a certificate of editing, but a signed letter from a colleague who is a native speaker of English is acceptable. Note the journal has arranged a number of discounts for authors

using professional language editing services
(<https://royalsociety.org/journals/authors/benefits/language-editing/>).

===PREPARING YOUR REVISION IN SCHOLARONE===

<https://royalsociety.org/journals/authors/author-guidelines/#supplementary-material> to include a suitable title and informative caption. An example of appropriate titling and captioning may be found at https://figshare.com/articles/Table_S2_from_Is_there_a_trade-

off_between_peak_performance_and_performance_breadth_across_temperatures_for_aerobic_sc
ope_in_teleost_fishes_/3843624.

Author's Response to Decision Letter for (RSOS-201229.R0)

See Appendix B.

Decision letter (RSOS-201229.R1)

Dear Dr Amos,

It is a pleasure to accept your manuscript entitled "Correlated and geographically predictable Neanderthal and Denisovan legacies are difficult to reconcile with a simple model based on inter-breeding." in its current form for publication in Royal Society Open Science. The comments of the reviewer(s) who reviewed your manuscript are included at the foot of this letter.

on behalf of Professor Andrés Ruiz-Linares (Associate Editor) and Kevin Padian (Subject Editor)
openscience@royalsociety.org

Associate Editor Comments to Author (Professor Andrés Ruiz-Linares):
Associate Editor
Comments to the Author:
Many thanks for the changes made to the manuscript.

Appendix A

Overall this paper utilized a five-populations allele sharing statistic to characterize the possible correlations i) between Denisovan and Neandertal introgression in worldwide populations, ii) between archaic introgression and geography. The author tested these correlations via simulated data, publicly available 1000Genomes and SGP datasets. The results presented in this paper are intriguing, though the results can be presented in a clearer way and interpreted more carefully (see comments below).

Major Comments:

1. The major conclusion of this paper relies on interpreting the *nd10* results. The author explains *nd10* in the first paragraph of page 6. By introducing five taxa in the paragraph (in contrast to four taxa in D statistics), I think it adds difficulty for readers to understand. It would be good to present *nd10* statistics in a simply one-line formula to make it easy to understand what *nd10* has been calculated.
2. Line 159, is figure 3 a zoom-in plot of figure 1? I am very confused because in the legend it says these are *nd10* values from Mallick et al 2020. It should be better explained.
3. The author has mentioned repeated in the paper that the archaic legacy size is stable and described *nd10* calculation for each chromosome in line 134-146. However there is zero plot on per-chromosome *nd10* results in this paper...
e.g. in Abstract: "Simulations confirm that, once created, legacy size is extremely stable: it may reduce through admixture with lower legacy populations but cannot increase detectably through neutral drift".
In Discussion: "The key issues are the consistency of signal across all chromosomes, which indicates that global variation in legacy size is not generated by either drift or selection".
4. In general the statements on mutation rate reads very confusing to me. It reads okay in the introduction part serving as background. But I do not see that it fits in in the discussion part. If the author insists to keep these statements, please make these statements read narrative and coherent. Also it is important to distinguish results, interpretations and speculations in clear language in the Discussion. Some sentences are mixed with speculations without clear word distinguishing results and speculations.

Minor Comments:

1. Line 102-104, to XXBAA, XXABA, XXBBA more understandable, it would be good to have a half sentence or a sentence to explain, e.g. XXBAA representing Denisovan and Chimpanzee alleles agree
2. Line 111-113, this sentence corresponds well to the results presented in figure 1. But how *nd10* was calculated for the rest of figures still needs to be explained – with or without conditioning on the African state and the use of transitions or transversions?
3. Line 189, "*nd10*_{NEA} is always constant across all non-African populations". I do not see the result for this. The legend of figure 7 says all values line

between 0.0155 and 0.0165. It would be good to have a panel in parallel in figure 7 showing $nd10_{NEA}$ values.

4. line 200, could the result be replicated with F_{st} , instead of AFD?
5. line 500, 'distance' typo
6. Figure 7, better to illustrate x-axis though there are some texts in the legend.

Appendix B

Dear Dr Amos

The Editors assigned to your paper RSOS-201229 "Correlated and geographically predictable Neanderthal and Denisovan legacies are difficult to reconcile with a simple model based on inter-breeding." have now received comments from reviewers and would like you to revise the paper in accordance with the reviewer comments and any comments from the Editors. Please note this decision does not guarantee eventual acceptance.

Please submit your revised manuscript and required files (see below) no later than 21 days from today's (ie 02-Mar-2021) date. Note: the ScholarOne system will 'lock' if submission of the revision is attempted 21 or more days after the deadline. If you do not think you will be able to meet this deadline please contact the editorial office immediately.

Subject Editor Comments to Author (Professor Kevin Padian):

Thank you for your submission. As you will see, the reviewers had some concerns that likely will need to be addressed with a bit more analysis. As such, if you need more time than the three weeks we customarily allot to major revisions, please let the editorial office know. Your resubmission may be returned to some reviewers for further assessment, so please take care to respond to their concerns. Best wishes.

Associate Editor Comments to Author (Professor Matthew Collins):

Comments to the Author:

Dear Prof Amos

I must apologize for the time it has taken (7 months) to review this MS which stemmed from challenge of finding reviewers for it, and then a number of reviewers who failed to provide reviews. Indeed your paper has been the most challenging to review of any that I have edited.

Response: this does not surprise me in the least, sorry!

Overall I think that the Abstract is slightly overselling the paper. Otherwise I see the comments as addressable by a minor review. The most significant issues to address are those raised by Reviewer #1 and regard the use of nd10.

Response: I have looked for statements that appear too strong and watered them down as much as I feel able. Also, I believe the additional work and results reflected in this revision add considerable extra support for my original conclusions, so I hope the abstract now seems balanced better with respect to the analyses presented.

Reviewer #1

In his manuscript "Correlated and geographically predictable Neanderthal and Denisovan legacies are difficult to reconcile with a simple model based on inter-breeding" the author cautions for the presence of artefacts or issues previously unaccounted for when describing the otherwise commonly accepted interaction between AMH and Neanderthal or Denisova.

I am not a native speaker, but I have some practical concerns on the style, since the manuscript is written almost as an essay rather than as a scientific article, and I think it would benefit from a small restructuring, to highlight better what are the 1) Problem 1; 2) Problem 2; 3) Problem 3 etc.. and to explain in a schematic way how the analyses proposed here address those problems and perhaps elucidate a more plausible alternative model which should be adopted instead.

Response: I have added extra text to the introduction that gives more background to why mutation rate might be expected to vary and also added a point-by-point set of the main predictions being tested. I hope this makes the text clearer.

For example it is not entirely clear whether the author is arguing
a) against the idea of Neanderthal and Denisova archaic introgressions OoA or
b) the current space/time models we have to reconstruct those events or
c) the alleged poor performance of nd10NEA and nd10DEN to separate the two archaic contributions or
d) a mix of the previous three points.

Response: I hope the revised text makes this clearer. The over-arching idea is to test two alternative models that can explain signals currently interpreted as evidence of introgression: genuine introgression and mutation slowdown out of Africa such the non-Africans appear closer to archaics compared with Africans. The various tests I perform are somewhat dependent on the performance of the admixture statistics being used, and my analysis uncovers some shortfalls, which are then discussed. This includes a new finding that 'introgression' is also detected from three great apes!

From my understanding of the author's concerns, essentially the two main points are:
1) Allele sharing stats calculated using nd10NEA and nd10DEN correlate more than would be expected by chance (it would be nice to see some correlation tests rather than just scatter plots) and

Response: I have added regression statistics and, where I feel it is appropriate, linear regression lines.

2) There is a geographic pattern (higher archaic legacy with higher distance OoA) which is not expected under either Neanderthal or Denisova introgression models and

Response: Yes! Though I now make it clearer that the simulations do show how such trends might be generated.

3) additionally, in the discussion, the author advocates for putative role of reduced mutation rate OoA.

Response: Partly, and I hope I now make this much clearer. If signals of introgression are not due to inter-breeding, they must be due to mutation rate variation: this is the only way to generate asymmetric base-sharing (non-zero D). My previous work has led me to conclude that heterozygosity modulates mutation rate. Across global human populations, heterozygosity declines approximately linearly with distance from Africa ($r^2 \sim 90\%$!, see Prugnolle et al.). If heterozygosity does modulate mutation rate, we expect

a parallel trend with introgression statistics like D , and this is exactly what we find. So, in short, if legacies are much smaller than currently thought or even do not exist, the primary requirement is for a lower mutation rate outside Africa. One model that seems capable of producing this is if heterogosity is mutagenic, and this is supported by the geographic trends in D and f_4 in the data.

My objections to the points above are:

1) Given the small number of archaic samples available, the nd_{10} approach is by no means "water-tight", since variants present in the available Denisova but not in the available Neanderthal genomes (and vice versa) are not granted to be "Denisova-specific" or "Neanderthal specific", and could instead fall on the shared Neanderthal-Denisova stem. Under this assumption (the author could prove me wrong with additional tests), the correlation observed between nd_{10NEA} and nd_{10DEN} could simply be the reflection of the contribution to either statistics of the shared Nea-Deni alleles. That said, if the author has concerns with this particular statistic (nd_{10}), I don't see why he couldn't simply adopt one of the many others available (including those that deal with haplotypes, rather than just single sites) to test the otherwise accepted scenario;

Response: This is an extremely important point and is one that has been a concern to me for a long time (and should be vastly more of a concern to others working in this field than is usually admitted). The fundamental problem is that all of the current measures are flawed, some deeply so, and this is an important conclusion from the work presented here. I now increase the emphasis on this in the revised text and have included a discussion. To address the Referee's comments:

1. **New analyses.** I have now added a new analysis based on a second very large dataset that uses the f_4 statistic, a close relative of D . I obtain very similar results. Indeed, the correlations between f_4 and distance from Africa and between comparable f_4 statistics for the two archaics are even stronger than with nd (or D).
2. **Adding more simulations.** Additional simulations and a fuller discussion of the original simulations presented both show that spillover of signal is rather modest. I also note that any given large value for one archaic is often linked to contrasting values in the second. For example, nd_{NEA} is around 0.025 in all non-American populations that are furthest from Africa, yet some have nd_{NEA} values around 0.05 and others 0.005. This shows clearly that in real data the problem of spillover, though likely present, is not large enough to drive the correlations.
3. **Simulating population samples for archaics.** I also tried running simulations that sampled 20 individuals from each archaic. The results were (of course!) identical, in that on average an allele is always sampled in proportion to its underlying frequency.
4. **Haplotypes.** I looked into using one of the methods based on haplotypes. The most easily accessible and best described version is the one used by Skov et al. However, reading their methods it become very apparent that they are effectively identifying bases that contribute to nd and then using a Hidden Markov Model to identify clustering. In this sense the method suffers all the issues inherent in nd (assuming back mutations are extremely rare, when there are 250,000 triallelic sites across the genome, that mutation rate is homogeneous when it varies massively by region and based type etc.) but in addition requires assumptions about mutational non-independence / a lack of correlation between mutation rate and recombination rate etc. etc.

2) Geographic pattern: given that both in real data and in the simulations provided here one of the populations further from Africa (Papua and to a lesser extent East Asians) have a boost of Denisova introgression, in light of the effect explained in the previous point it is not surprising that also the nd_{10NEA} experience such a correlation. Indeed,

while the amount of true Neanderthal component may remain the same across all Eurasia, the latent Denisova component that diffuses westward from Papua/East Asia through isolation by distance can mimic a higher Neanderthal contribution.

Response: I thank the Reviewer for this very interesting suggestion. However, the Denisovan fraction in PNG drops dramatically both between PNG and the mainland and even between different Papuan population groups. This sudden drop indicates clearly that gene flow between the PNG groups with high legacy and other groups has been minimal. Indeed, Denisovan legacies outside Australasia are often reported to be near-zero. Conversely, the high Neanderthal legacy seen in PNG is arguably not nearly high enough if there has been a significant spill-over from Denisovans: we would expect to see a jump in estimated Neanderthal legacy size comparable to the large jump seen for Denisovans and we do not. This actually adds a useful additional argument that nd10 is performing reasonably well!

3) I didn't entirely get the role of reduced mutation rate in the overall criticism offered by the author, however I think that invoking longer generation times OoA as an alternative/concurrent explanation may result in a similar effect, and perhaps into a more readily acceptable one by the non expert readership.

Response: I have tried to make the text much clearer by expanding my discussion of this. The key point is that there are only two mechanisms capable of generating non-zero D statistics, introgression and mutation rate variation sufficient to cause appreciable differences in branch lengths. Changes in generation length can create effects that in some ways mimic changes in mutation rate. However, the observed pattern requires that that generation times got progressively longer as humans dispersed away from Africa (equivalent to the mutation rate getting progressively lower). If anything, under stresses such as might be experienced by populations expanding at low densities into new habitats, most species that have been studied show a shortening of generation time. In addition, the new analyses point to effects that apply variably across the genome, something that cannot be explained by generation length alone.

Minor point:

Please include color legends in the figures for ease of reading and if possible also pearson r and p-values when you refer to correlations.

Response: Done as requested.

Review #2

This paper concerns geographic trends of archaic (Neanderthal and Denisovan) legacies in modern human genomes, analyzed based on data from several sources and simple models of migration and introgression, as well as computer simulations.

The author finds a number of inconsistencies of data compared to theoretical models. One of them is the complete reversal of correlations of nd10(NEA) vs. nd10(DEN) (Figure 1), depending on whether they are conditional on absence of archaic admixture in subsaharan Africa (predominant hypothesis) or they are unconditional. On the other hand, a nd10(NEA) vs. nd10(DEN) correlation among Africans follows a complicated pattern (Figure 2) although it is not clear how reverse-migration flow might contribute to it. Influence of selection is less likely, since the principal trends are shared in all chromosomes. Another interesting result is summarized in Fig.5 and it shows that the residuals of Neanderthal nd10 statistics corrected for land-distance from Africa, follow an East-West increase of Neanderthal legacy, which is partly inconsistent with the geographic range of Neanderthals (highest residuals in China).

A Discussion follows, which finds these findings in contradiction to any simple models of migration-related introgressions. Data might be found consistent with a model assuming that mutation rate is positively correlated with heterozygosity, while the latter decreases with the distance from Africa. However this relationship, once proposed by Amos, is by

his own admission, not commonly accepted. One of the attractive features of the paper is a simulation study, which puts together the major tenets (however, please see further). The most general conclusion of the work is that if the introgressions are real (the Neanderthal one seemingly founded on more extensive evidence), then further effort is needed to understand their nature.

I found one aspect of the paper which needs some improvement. The Discussion of simulation results might be made clearer, for example by making direct references to color-coded lines plotted in Figure 7 (referring to various cases which correspond or do not correspond to data). This reader seems somewhat lost without this and it can be fixed with relatively simple text alterations.

Response: I thank the reviewer for this constructive comment. I have now extended the simulation section in a way that I hope improves clarity and addresses both these concerns and points raised by reviewer 1. More simulations are included.

Figure 5: Color scale seems more like yellow - brown - purple -violet, than what is claimed in the legend (yellow - brown - mauve - blue). I would use a brighter color scale, if possible.

Response: The reviewers' comments together caused me to reconsider the point of this figure and I concluded that it was minimal. After all, if residuals vary along a regression the regression itself is not valid! Consequently, I have removed this figure.

Reviewer #3:

Overall this paper utilized a five-populations allele sharing statistic to characterize the possible correlations

- i) between Denisovan and Neandertal introgression in worldwide populations,
- ii) between archaic introgression and geography.

The author tested these correlations via simulated data, publicly available 1000Genomes and SGDP datasets. The results presented in this paper are intriguing, though the results can be presented in a clearer way and interpreted more carefully (see comments below).

Major Comments:

1. The major conclusion of this paper relies on interpreting the nd_{10} results. The author explains nd_{10} in the first paragraph of page 6. By introducing five taxa in the paragraph (in contrast to four taxa in D statistics), I think it adds difficulty for readers to understand. It would be good to present nd_{10} statistics in a simply one-line formula to make it easy to understand what nd_{10} has been calculated.

Response: I have reworded this section to make much clearer and added a simple equation to the methods section.

2. Line 159, is figure 3 a zoom-in plot of figure 1? I am very confused because in the legend it says these are nd_{10} values from Mallick et al 2020. It should be better explained.

Response: The figure is indeed a 'zoom in'. The legend says clearly "This figure is the same as Figure 2 except that African samples (apart from four North African samples) and samples with very high nd_{DEN} values (Australia and Papua New Guinea) are excluded." I am not sure how to make this any clearer.

3. The author has mentioned repeatedly in the paper that the archaic legacy size is stable and described nd_{10} calculation for each chromosome in line 134-146. However there is zero plot on per-chromosome nd_{10} results in this paper... e.g. in the Abstract: "Simulations confirm that, once created, legacy size is extremely stable: it may reduce through admixture with lower legacy populations but cannot increase detectably through neutral drift".

Response: I have expanded to simulation section to include more scenarios, including plots of what is now called nd_{NEA} , showing how stable it is. I hope this is sufficient to address these concerns.

In Discussion: "The key issues are the consistency of signal across all chromosomes, which indicates that global variation in legacy size is not generated by either drift or selection".

4. In general I found the statements on mutation rate to be confusing. They read clearly in the introduction which serves as background but fit less well in the discussion. If the author insists on keeping these statements, please check their consistency between sections. Also it is important to distinguish between results, interpretations and speculations in the Discussion. Some sentences combine interpretation and speculation, and it would be useful to wordsmith these to discriminate between results and speculation.

Response: I hope everything is clearer now, though without guidance as to exactly what the problems are it is difficult to be sure.

Minor Comments:

1. Line 102-104, to XXBAA, XXABA, XXBBA more understandable, it would be good to have a half sentence or a sentence to explain, e.g. representing Denisovan and Chimpanzee alleles agree

Response: I have added extra text along these lines.

2. Line 111-113, this sentence corresponds well to the results presented in figure 1. But how nd_{10} was calculated for the rest of figures still needs to be explained – with or without conditioning on the African state and the use of transitions or transversions?

Response: I have added an equation to the method section and an explicit statement of how nd was calculated.

3. Line 189, " nd_{10NEA} is always constant across all non-African populations". I do not see the result for this. The legend of figure 7 says all values line between 0.0155 and 0.0165. It would be good to have a panel in parallel in figure 7 showing nd_{10NEA} values.

Response: I have added the requested figure.

4. line 200, could the result be replicated with F_{st} , instead of A_{FD} ? 5. line 500, 'distance' typo

Response: corrected. I also repeated the analyses with F_{st} and obtained essentially identical results.

6. Figure 7, better to illustrate x-axis though there are some texts in the legend.

Response: I am not sure I understand this bit, sorry!